# Challenges in global climate models to represent cloud response to aerosols: insights from volcanic eruptions

Yu Wang [1] ✉, David Neubauer [2], Ying Chen [3], George Jordan [4], Florent Malavelle[5], Tianle Yuan[6,7], Daniel Partridge [8], Paul Field[5,9], Hao Wang[10], Minghuai Wang [10], Martine Michou[11], Pierre Nabat [11], Anton Laakso[12], Gunnar Myhre [13] & Ulrike Lohmann [2]

Aerosol-cloud interactions (ACI) remain a major source of climate uncertainty due to missing large-scale observational constraints. Such a constraint, with global cloud representativeness, has recently been developed based on the Holuhraun-2014 volcanic eruption from machine learning with satellite observations. Here, we confront this large-scale observational constraint against six diverse global climate models to advance our understanding of ACI simulation uncertainty. We show that marine liquid cloud optical depth responses to aerosols are reasonably well simulated, although through compensating errors. However, all models largely underestimate cloud cover responses to aerosols, with five of them outside the 90% confidence level. This persistent bias remains despite tuning five distinct cloud schemes and testing various key cloud processes. Such bias in cloud cover response is a major driver of simulation uncertainty in ACI cooling and needs to be addressed urgently to improve climate projections and estimations of climate sensitivity.

The global mean near-surface temperature in 2024 was a record breaking +1.55 °C above pre-industrial levels, according to WMO, temporarily overshooting the +1.5 °C threshold. Under current emission trajectories, a warming well above +2 °C is expected by 2100, driving devastating hazards[1,2]. Accurate climate projections are needed to guide policymaking for decarbonisation strategies and climate interventions to minimise losses from subsequent climate and weather extremes[1,2].

A series of Intergovernmental Panel on Climate Change (IPCC) reports has identified large uncertainties in climate assessments[1,2]. One major source of uncertainty is aerosols, tiny particles suspended in the atmosphere, and their impacts on clouds[1,2]. Modern-day global observations (e.g., satellite, aircraft, in-situ) have improved constraining simulations of aerosol and cloud in terms of the present-day climate (e.g., refs. [3,4]). However, how clouds respond to aerosol perturbations (usually quantified as the susceptibility of cloud properties to changes in aerosol, "dlnCloud/dlnAerosol") is still poorly constrained[5–7]. Cloud susceptibility is critical for climate projections, since "Future clouds = Current cloud state + Exp(dlnCloud/dlnAerosol × ΔlnAerosol) + Feedback of clouds to climate change". The long-standing challenge for constraining cloud susceptibility is the lack of large-scale observational constraints to calibrate global climate

[1]School of GeoSciences, University of Edinburgh, Edinburgh, UK. [2]Institute for Atmospheric and Climate Science, ETH Zürich, Zürich, Switzerland. [3]School of Geography, Earth and Environmental Sciences, University of Birmingham, Birmingham, UK. [4]Met Office Hadley Centre, Exeter, UK. [5]Met Office, Exeter, UK. [6]Goddard Earth Sciences Technology and Research (GESTAR) II, University of Maryland, Baltimore, MD, USA. [7]Sciences and Exploration Directorate, Goddard Space Flight Center, Greenbelt, MD, USA. [8]College of Engineering, Mathematics, and Physical Sciences, University of Exeter, Exeter, UK. [9]School of Earth and Environment, University of Leeds, Leeds, UK. [10]School of Atmospheric Sciences, Nanjing University, Nanjing, China. [11]Météo-France, CNRS, Univ. Toulouse CNRM, Toulouse, France. [12]Finnish Meteorological Institute, Kuopio, Finland. [13]CICERO Center for International Climate Research, Oslo, Norway. ✉e-mail: y.w@ed.ac.uk

models (GCMs), leaving discrepancies of more than a factor of 10 that have existed for decades amongst GCMs[8]. These large discrepancies in cloud susceptibility can propagate significant uncertainties in climate projections across future scenarios, even with similar current states of clouds and aerosols. Therefore, constraining cloud processes and susceptibility is critical for reducing climate uncertainty[5,8,9].

Aerosols can influence cloud formation by serving as cloud condensation nuclei (CCN) and ice-nucleating particles[1]. In this study, we focus on marine warm clouds given that their net effect on radiative forcing is the largest and their likeliness to be perturbed by human emissions due to their low altitude[10]. Increased aerosol concentration results in more CCN and more, yet smaller, cloud droplets (effective droplet radius ($R_e$) decrease), making clouds brighter (Twomey effect)[11]. In response, clouds tend to adjust their liquid water path (LWP) and liquid cloud cover (LCC) through complex cloud microphysical processes. Two competing effects determine the changes of LWP and LCC to aerosol perturbations. On the one hand, numerous smaller cloud droplets could suppress precipitation, leading to an increase in LWP and LCC, and prolonging cloud lifetime (Albrecht effect)[12,13]. On the other hand, numerous smaller cloud droplets could increase cloud-top entrainment of dry air, leading to droplet evaporation and hence decreasing LWP and LCC (entrainment effect)[14]. Both the Twomey effect and the larger LWP result in a negative forcing via optically thicker and brighter clouds, resulting in more reflection of sunlight and cooling. In addition, a larger LCC has been found to contribute to a large negative forcing[6,7].

To date, the Twomey effect[11] has been well documented in multiple lines of evidence, including modelling (e.g., refs. 15,16) and observations (e.g., refs. 6,7,17–21). However, evidence for the cloud Albrecht effect, including LWP and LCC adjustments, is much less clear, with signals in the range from positive to negative reported[17,21–29]. A major challenge is disentangling aerosol impacts on clouds from confounding meteorological co-variability, especially on a climate-relevant scale of thousands of km where meteorological conditions can differ spatially and temporally[21]. In 2014, the Icelandic effusive eruption known as Holuhraun resulted in an aerosol plume over the North Atlantic for months, affecting clouds with a regime spectrum analogous to global cloud regimes[21,30]. This opportunistic experiment has been proposed to serve as a benchmark case for studying ACI at a climate-relevant scale[21,30]. The challenge of confounding meteorology has been addressed in a recent work[6], using machine-learning-based counterfactual clouds without volcanic influence to contrast real-world observations (i.e. clouds with volcanic influence). The detected ACI signals effectively minimise the noise from satellite observational uncertainty, by aggregating over an extensive region spanning millions of square kilometres[6]. By applying this machine-learning approach in Hawaii trade wind region, large ACI signals were detected in the downstream of volcanic aerosol plume while negligible signals in the upstream[7]; suggesting this methodology is able to estimate cloud susceptibility at large scale from satellite observations, making it possible to calibrate cloud susceptibility in GCMs.

Now, this study leverages the observational constraint of cloud susceptibility at a climate-relevant scale to challenge a group of six diverse GCMs with distinct cloud schemes, the group chosen as they well represent the uncertainty range of current state-of-the-art GCMs[8]. We found that all models significantly underestimate cloud cover response to the aerosol perturbation, although they simulate cloud optical depth response reasonably well, but with compensating errors. This multi-model intercomparison pinpointed key ACI uncertainties. Here, we echo Malavelle et al.[21] to call for the wider climate community takes the Holuhraun-2014 eruption as an anchor case, and stress validation of cloud susceptibilities in addition to cloud properties to decompose and unravel the largest uncertainty in climate radiative forcing, i.e. ACI simulation, therefore pointing towards a vital direction to improve future climate simulations.

## Results

### Multi-model ACI structural uncertainty

The structural uncertainty in ACI simulations refers to the uncertainty arising from limitations in the representations of physical processes in models[8]. Here we use the large-scale observational constraint of cloud susceptibility from Holuhraun-2014[6], to challenge six GCMs and uncover their structural uncertainty of ACI. These GCMs include ECHAM6.3-HAM2.3[31], CESM2.1.0[32], UKESM1[33], CNRM-ESM2-1[34], ECHAM6.3-SALSA2.0[35], CAM5.3-Oslo[36]. The discussion and evaluation of cloud state simulations are consistent with a previous study by Malavelle et al.[21], see details in the Supplementary Discussion Section S1. In contrast to the widely used method of evaluating cloud and aerosol states, here we re-evaluate the performance of GCMs using cloud susceptibilities as a proxy for ACI. Cloud susceptibilities are defined as changes in cloud properties (i.e., $R_e$, LWP) and LCC in response to volcanic aerosol-induced changes in cloud droplet number concentration, $N_d$ (i.e., $-dlnR_e/dlnN_d$, $dlnLWP/dlnN_d$, and $dlnLCC/dlnN_d$), using paired GCM simulations with and without Holuhraun-2014 volcanic emissions (see Methods). Together, these cloud susceptibilities determine the change in cloud radiative effect.

Figure 1a displays the results of the evaluation of GCMs overlaid on violin plots illustrating the observational ACI constraints[6]. These violin plots represent the probability distributions of each cloud susceptibility with a 90% confidence interval, i.e., "very likely" as defined in the IPCC uncertainty guideline[37]. The best estimates (median values [90% confidence interval]) are: $-dlnR_e/dlnN_d = 0.37$ [0.16–0.92], $dlnLWP/dlnN_d = 0.02$ [−0.18–0.21], $dlnCOD/dlnN_d = 0.39$ [0.11–0.96], and $dlnLCC/dlnN_d = 0.42$ [0.09–1.06], where COD stands for cloud optical depth. The Twomey effect, indicated by $-dlnR_e/dlnN_d$, is relatively well simulated in six GCMs, falling within the 90% likelihood of the observational constraint, although they are slightly underestimated (below the 50th percentile), ranging from 0.15 in ECHAM6.3-HAM2.3/UKESM1 to 0.32 in CNRM-ESM2-1.

Regarding the liquid water path adjustments ($dlnLWP/dlnN_d$), all models except for CNRM-ESM2-1 overestimate the LWP response to changes in $N_d$ compared to the observational constraint. Four GCMs fall within the observational uncertainty range, from the 50th percentile for CNRM-ESM2-1 to the 90th percentile for CAM5.3-Oslo. Two GCMs, ECHAM6.3-HAM2.3 and ECHAM6.3-SALSA2.0, show excessive LWP adjustments, far exceeding the observational constraints. These overly strong LWP adjustments in some of the GCMs compared to machine-learning-based observational constraints[6] are in line with Malavelle et al.[21], who compared four GCMs with climatological anomalies from satellite observations[21]. However, we further show that this overestimation of LWP adjustment in GCMs is balanced by the underestimation of the Twomey effect in all six GCMs to give a reasonable value of cloud optical depth response to ACI.

The COD (an indicator of albedo) response can be obtained by combining the Twomey effect and LWP adjustment (i.e., $dlnCOD/dlnN_d = -dlnR_e/dlnN_d + dlnLWP/dlnN_d$)[8]. All models simulate the COD response well within 25–75th percentiles of the observational constraint. Although, except for CNRM-ESM2-1, five GCMs do so by compensating between low biases in the Twomey effect and high biases in LWP adjustment. An exception is CNRM-ESM2-1, which excels in simulating both the Twomey effect and the LWP adjustments in best agreement with observations. The $dlnCOD/dlnN_d$ in ECHAM6.3-HAM2.3 and ECHAM6.3-SALSA2.0 is higher than the best estimate, primarily driven by the excessively strong LWP adjustment. Meanwhile, the low $dlnCOD/dlnN_d$ in UKESM1 is driven by an overly weak Twomey effect.

We find that the susceptibility of cloud cover is largely underestimated in all six GCMs. Almost all GCMs fall out of the 90% confidence interval of observational constraints, with only ECHAM6.3-SALSA2.0 touching the lower bound, indicating "very unlikely" that

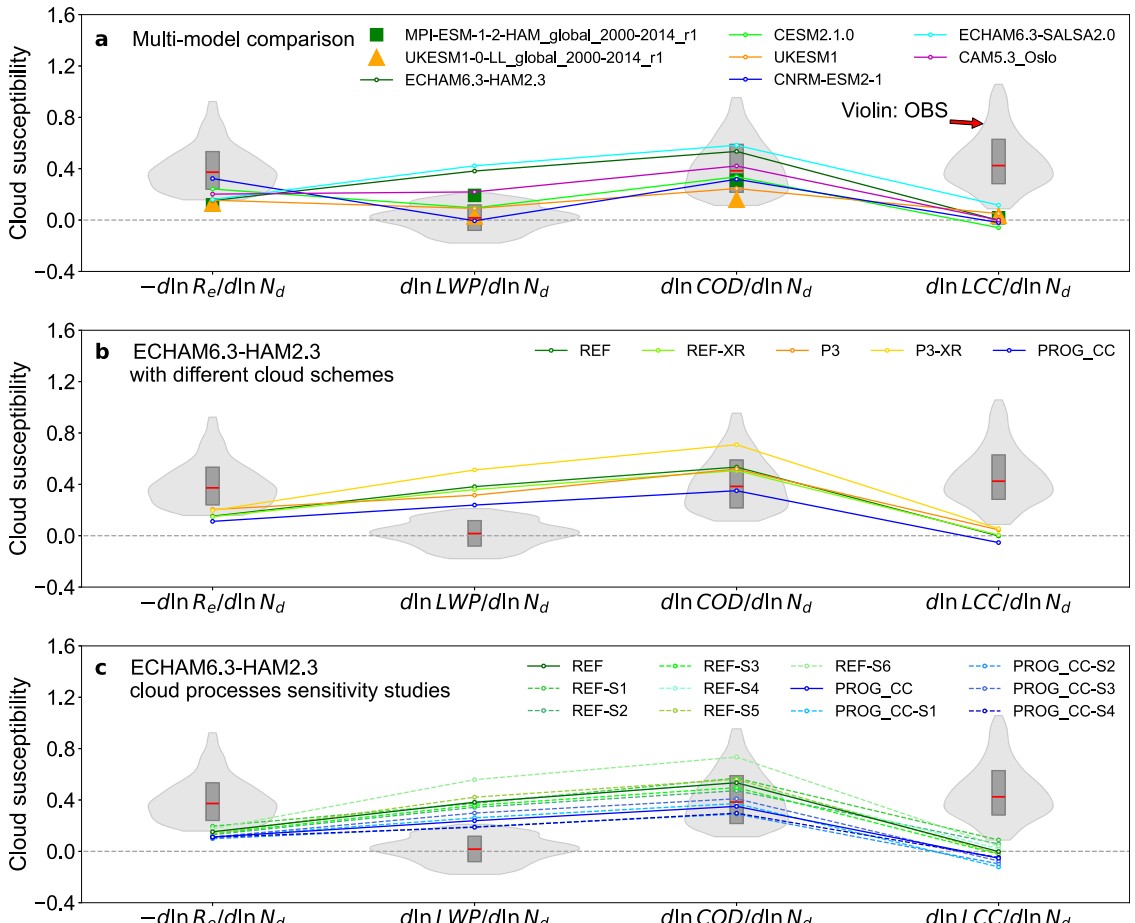

**Fig. 1 | Intercomparison of liquid cloud properties response to aerosol perturbations (i.e., cloud susceptibilities) from volcanic degassing at Holuhraun, Iceland (October 2014), based on multiple model simulations with varying cloud schemes and process sensitivities, alongside observational constraints[6].** Cloud susceptibilities (-dlnRe/dlnNd, dlnLWP/dlnNd, dlnCOD/dlnNd, and dlnLCC/dlnNd) are changes in cloud properties (Re cloud droplet effective radius, LWP liquid water path, COD cloud optical depth, LCC liquid cloud cover) in response to changes in droplet number concentration, $N_d$. **a** Six general circulation models (GCMs) from this study, representative of the Holuhraun region behaviour, and two GCMs from the Coupled Model Intercomparison Project Phase 6 (CMIP6), representative of the global behaviour. **b** ECHAM6.3-HAM2.3 model with varied cloud schemes. **c** ECHAM6.3-HAM2.3 model (default cloud scheme) with cloud process sensitivities. The violin plots represent 90% confidence intervals of machine-learning-based observational constraints[6], with the inner box showing the 25th, 50th, and 75th percentiles. Note: Liquid cloud cover is not available in CMIP6, so total cloud cover is used to calculate dlnLCC/dlnNd for MPI-ESM-1-2-HAM and UKESM1-0-LL in panel a.

## Table 1 | Model treatments

| Model name | Aerosol | CCN activation | Stratiform cloud microphysics | Cloud cover scheme |
|---|---|---|---|---|
| ECHAM6.3-HAM2.3 (REF) | M7[74,75] | ARG[76] | LN[46]: 2-mom | Sundqvist[40] |
| CESM2.1.0 | MAM4[77] | ARG[76] | MG2[78]: 2-mom | CLUBB[38,39] |
| UKESM1 | GLOMAP-mode[79] | ARG[76] | WB99[80]: 1-mom | PC2[41] |
| CNRM-ESM2-1 | TACTIC_v2[81] | Menon02[82] | Lopez02[42]: 1-mom | Roehrig20[34] |
| ECHAM6.3-SALSA2.0 | SALSA2.0[35] | ARG[76] | LN[46]: 2-mom | Sundqvist[40] |
| CAM5.3-Oslo | OsloAero5.3[36] | ARG[76] | MG1.5[83]: 2-mom | Park14[43] |

current state-of-the-art climate models are able to capture the aerosol-induced cloud cover increase. Liquid cloud cover is usually parameterised in terms of relative humidity (RH) and liquid water content in CESM2.1.0[38,39] or only a function of RH in the other five GCMs[40-43] (Table 1 in Methods). This systematic bias indicates a lack of connection between aerosols and cloud cover parameterisations in GCMs. Further improvements in representations in entrainment or rain formation could improve cloud lifetime[44,45] and thus cloud cover responses to aerosol changes in GCMs.

## Multi-scheme ACI structural uncertainty

Although the cloud cover in models is diagnosed by grid-mean RH, it remains closely linked to cloud microphysical processes through the partitioning of water in the vapour, liquid, and ice phases. Thus, the chosen cloud schemes in the model are expected to influence cloud susceptibilities, causing structural uncertainty of ACI. To further understand this issue, we select ECHAM6.3-HAM2.3 as a showcase model to explore whether ACI uncertainty, especially for cloud cover, can be reduced by implementing different cloud schemes. ECHAM6.3-

**Table 2 | Perturbed parameters in sensitivity studies in ECHAM6.3-HAM2.3**

| Sentsitiy runs | Perturbed parameter |
|---|---|
| REF | Default diagnostic scheme |
| REF-S1 | Reducing autoconversion rate (ccraut) in stratiform clouds from 5 (default value) to 1 → Delayed rain formation in stratiform clouds |
| REF-S2 | Reducing threshold RH $\eta_{crit}$ by applying a scaling factor of 0.8, only over the ocean → more marine clouds |
| REF-S3 | Increasing evaporation of raindrops to 200% of default evaporation rate → Increase RH and therefore cloud cover |
| REF-S4 | Reducing entrainment rate for shallow convective clouds (entrscv) from 0.003 to 0.001 → Drying the boundary layer and reducing cloud cover |
| REF-S5 | Reducing fraction of overshooting of shallow convection (cmfctop) from 0.2 to 0.1 → Keeping more moisture in the planetary boundary layer |
| REF-S6 | Reducing autoconversion rate for convective clouds (cprcon) from 0.0009 to 0.00002 → Delayed rain formation in convective clouds |
| PROG_CC | Default prognostic scheme |
| PROG_CC-S1 | Reducing scaling factor for cloud formation by mixing of detrained air with environmental cloud-free air (tuningDetrainedMixing) from 0.8 (default value) to 0.6 → Reducing cloud formation |
| PROG_CC-S2 | Reducing scaling factor for entrainment rate for shallow convection (tuningConvEntrShallow) from 0.003 (default value) to 0.0003 → Drying the boundary layer and reducing cloud cover |
| PROG_CC-S3 | Increasing scaling factor for turbulent mixing (tuningTurbulentMixing) from 0.0 (default value) to 1.0 → Affect cloud formation |
| PROG_CC-S4 | Reducing critical RH at the surface above which large-scale cloud cover can form (tuningRHcritSurface) from 0.9 (default value) to 0.8 everywhere → more clouds |

HAM2.3 is chosen because its LCC adjustment is about in the middle of six GCMs, indicating a good representative (in line with Ghan et al.[8]) and because multiple cloud schemes (both diagnostic and prognostic ones) have been embedded in the model.

In total, five different cloud schemes have been implemented in the ECHAM6.3-HAM2.3 model to test various cloud microphysics and cover parameterisations. The default cloud schemes include a two-moment cloud microphysics scheme[46] and an RH-based cloud cover scheme[40], referred to as REF. One additional cloud microphysics scheme is implemented: Predicted Particle Properties scheme (P3)[47]. Next to the default RH-based cloud cover scheme, we implement the Xu and Randall cloud cover scheme (XR)[48] with REF and P3 microphysics schemes. In the XR scheme, the cloud cover is determined by both liquid water and RH[48]. In addition, we used the Prognostic cloud cover scheme (PROG_CC)[49], a Tiedtke-like scheme[50] similar to that in GFDL's CM4.0 Climate Model[51,52], which simulates cloud fraction prognostically based on in-cloud water vapour and convective activity and uses an updated cloud microphysics scheme. In total, we have five different cloud scheme setups in ECHAM6.3-HAM2.3, including REF, REF-XR, P3, P3-XR, and PROG_CC.

As shown in Fig. 1b, all five cloud schemes in ECHAM6.3-HAM2.3 demonstrate a reasonable simulation of the COD response. The reasonable COD response is again due to the compensation of biases in the Twomey effect and the LWP adjustment. The Twomey effect ($-d\ln R_e/d\ln N_d$) is underestimated, falling below the 25th percentile of the observational constraint, although the simulations with the P3 scheme perform slightly better than with the REF and PROG_CC schemes. The LWP adjustment ($d\ln LWP/d\ln N_d$) in all schemes exceeds the observational constraint, showing excessively strong signals, even though PROG_CC is much improved (close to the 90th percentile) compared to other schemes. As a result, the ensemble spans the range of the observed COD response ($d\ln COD/d\ln N_d$) within the 45–85th percentiles. Nevertheless, none of the cloud schemes reproduce the observed increase in cloud cover ($d\ln LCC/d\ln N_d$), even the cloud cover schemes that depend on liquid water (XR) or prognostic scheme (PROG_CC) still underestimate $d\ln LCC/d\ln N_d$, suggesting a major structural uncertainty in current cloud cover schemes. Further investigations of the cloud cover schemes and relevant cloud processes are imperative to clarify cloud cover response to aerosol changes and to inform model improvements.

## ACI parametric uncertainty
Similar to the cloud schemes, tuned cloud microphysical processes impact the partitioning of water in the vapour, liquid, and ice phases,

therefore impacting cloud cover and causing parametric uncertainty of ACI. To further explore the influence of individual cloud microphysical processes on cloud susceptibilities, we selected two distinct cloud schemes, REF (diagnostic cloud cover scheme) and PROG_CC (prognostic cloud cover scheme). Detailed parameter settings of the sensitivity studies are listed in Table 2 (Methods). Briefly, based on a previous ECHAM model tuning study[53], we tuned the parameters of key microphysical processes that influence cloud cover towards maximising cloud cover response to changes in aerosols[53]. For the default scheme (REF), the perturbed processes include autoconversion rates for stratiform and convective clouds, the RH threshold for cloud formation, entrainment rate, and the fraction of shallow convection into the free troposphere (details in Table 2). Similarly, we perturbed processes for the prognostic cloud cover schemes (PROG_CC), including the detrainment rate, entrainment rate, turbulence, and the RH threshold for large-scale cloud formation.

As shown in Fig. 1c, for the REF scheme, despite the perturbed simulations slightly increasing the cloud cover susceptibility compared to the default setup, as expected, the $d\ln LCC/d\ln N_d$ fell outside the range of the observational constraint. As a side effect, the perturbed runs simulated an even higher bias in LWP adjustment ($d\ln LWP/d\ln N_d$) than the default setting (except for REF-S2), whilst the Twomey effect ($-d\ln R_e/d\ln N_d$) is relatively insensitive, leading to a greater overestimation of the COD response (green hue lines). For the PROG_CC scheme (blue hue lines), perturbed runs only changed cloud cover responses and COD response slightly. The reasons for this are unclear, but the prognostic cloud cover scheme[49] has more degrees of freedom to respond to cloud processes compared to the default diagnostic scheme REF[46]. These dichotomous results further highlight the complexity of influencing the cloud cover response through cloud microphysical processes. Since different cloud schemes and models may provide various responses, further investigations with a more detailed evaluation of the results with ECHAM6.3-HAM2.3 and more models are required to address this issue.

## Global representativeness and robustness of the Holuhraun eruption
To evaluate the global representativeness of the Holuhraun-2014 eruption case, we compared its cloud regime distribution and cloud susceptibilities to global ones. Malavelle et al.[21] and Chen et al.[6] have shown that the Holuhraun-2014 eruption occurred in a region encompassing the full range of cloud regimes, whose distribution is closely matched with the global distribution observed by satellite from 2002 to 2014 (see Extended Data Fig. 1 in Chen et al.[6]). In addition, we

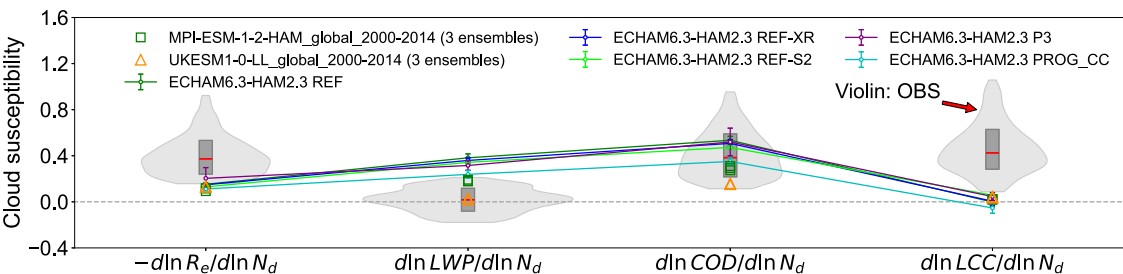

**Fig. 2 | Intercomparison of liquid cloud properties response to aerosol perturbations (i.e., cloud susceptibilities) from Holuhraun-2014 eruption in ECHAM6.3-HAM2.3 with different cloud schemes and sensitivities (each experiment with ten ensemble runs, ensemble mean values and one standard deviation are shown) and two models from the Coupled Model Intercomparison Project Phase 6 (MPI-ESM-1-2-HAM and UKESM1-0-LL, each with three ensemble runs).** Cloud susceptibilities (-dlnRe/dlnNd, dlnLWP/dlnNd, dlnCOD/dlnNd, and dlnLCC/dlnNd) are changes in cloud properties (Re cloud droplet effective radius, LWP liquid water path, COD cloud optical depth, LCC liquid cloud cover) in response to changes in droplet number concentration, Nd. The violin plots represent 90% confidence intervals of machine-learning-based observational constraints[6], with the inner box showing the 25th, 50th, and 75th percentiles. Note: the three ensemble runs for each CMIP6 model are all plotted, but they overlap with each other due to negligible differences.

assessed cloud susceptibilities at a global scale resulting from anthropogenic aerosol emissions throughout 2000–2014 in CMIP6 simulations (MPI-ESM-1-2-HAM and UKESM1-0-LL, equivalent to ECHAM2.3-HAM2.3 and UKESM1 in this study, respectively). We compare CMIP6 simulations of "historical" and "hist-piAer" experiments, representing with and without anthropogenic aerosol influence, respectively; details are given in the Methods. The susceptibilities of clouds to anthropogenic aerosol emissions at the global scale are consistent with the Holuhraun-2014 eruption in our GCMs simulations (Fig. 1a). The consistency in both cloud regimes distribution and cloud susceptibilities suggests that the Holuhraun-2014 eruption provides a valid and representative analogue for understanding global cloud responses to aerosol perturbations due to human emissions in recent years.

To strengthen robustness of our results, we quantified model uncertainty caused by the inherently chaotic nature of cloud dynamics and radiative effects through ensemble simulations. We used ECHAM6.3-HAM2.3 as a showcase model and explored uncertainty across five representative experiments, each reflecting distinct cloud schemes and process sensitivities, with ten ensemble simulations per experiment. We find only a small degree of uncertainty is introduced by model internal variability. For cloud cover, the ensemble uncertainty is a magnitude smaller than the gap between GCMs and the observation (Fig. 2). Furthermore, we demonstrate the robustness in global representativeness using ensembles in CMIP6 simulations (three ensembles for each). Again, a negligible difference is found between CMIP6 ensembles (Fig. 2). These well demonstrate the robustness of the key conclusions and its global representativeness in this study.

## Discussion

Our study highlights two critical needs to reduce ACI uncertainty in global climate models. Firstly, to improve simulation of processes relevant to COD response so that we get the "correct answer for the right reasons" for cloud albedo change, we must investigate aerosol and cloud microphysics schemes in more detail to better represent the Twomey effect and the LWP adjustment. Secondly, it is essential to further develop model schemes that better represent processes (e.g., entrainment, auto-conversion, etc.) for the cloud cover responses to aerosol changes (hence the changes in cloud droplet number). High-resolution large-eddy simulations that more explicitly simulate cloud microphysical processes, have the potential to simulate the cloud cover increase in response to aerosol changes[54]. A combination of machine learning and the results of high-resolution cloud-resolving models would be a plausible way forward for improving cloud parameterisations in global models to better represent the cloud cover's responses to aerosol changes[55].

Our study is not without limitation. The observational constraints may suffer from uncertainty in satellite observations[56], even though satellite retrievals are currently the only measure to provide continuous long-term cloud cover observations over a large scale. In future studies, using multiple satellites along with machine-learning development could further reduce uncertainty in the observations of cloud cover and cloud susceptibility to better constrain global climate models. The nonlinearity of cloud response to aerosol means that cloud susceptibilities can also depend on meteorology[7,57] and the strength of aerosol source compared to background[9,57,58]. For example, Gettelman et al.[57] performed sensitivity simulations of Holuhraun-2014 using a GCM and demonstrated that meteorological diversity could introduce up to 30% difference in ACI radiative forcing, and that the forcing is not proportional to the change in aerosol emissions. This suggests the need of more opportunistic experiment studies, such as IMO shipping emission reduction since 2020[59], together with the Holuhraun-2014 to have a holistic investigation of ACI in diverse global conditions. In addition, due to the complexity of clouds and climate systems, even a perfect simulation of cloud susceptibilities still cannot guarantee an unbiased estimate of ACI radiative forcing, this is because the efficiency in radiative effect of clouds varies significantly between GCMs; for example, CESM2 shows similar cloud susceptibilities to CNRM-ESM2-1 (Fig. 1a), but shows some of three times larger ACI radiative forcing[60]. A recent study highlighted that high equilibrium climate sensitivity (≥2.93 K) is needed to reproduce the trend in the observed Earth energy imbalance, where changes of clouds are critical for climate assessments but highly uncertain[61]. Therefore, radiative transfer processes of clouds and cloud feedback following ACI radiative forcing are also key to improving climate projections.

## Methods
### Model setup
The effusive volcanic eruptions at Holuhraun (Iceland, 2014) emitted tens to hundreds of kilotons of $SO_2$, creating a sulfate aerosol plume in the marine boundary layer which spread over the whole North Atlantic[21], providing an ideal natural experiment to investigate aerosol-cloud interactions. Malavelle and Yuan proposed an AeroCom multi-model intercomparison project (VolcACI)[62–64] to use this Holuhraun eruption as a benchmark case to study uncertainty in aerosol-cloud interaction simulations. In this study, we used five GCMs from the VolcACI project, based on available data for analysis, including UKESM1[33], CNRM-ESM2-1[34], ECHAM6.3-SALSA2.0[35], CAM5.3-Oslo[36,45], ECHAM6.3-HAM2.3[31]. In addition, CESM2.1.0[32,65] was added for this experiment following the VolcACI protocol.

The model setups and the simulations followed the VolcACI protocols[62]. Model treatments of aerosols and cloud schemes are

**Table 3 | Experiment information for the selected CMIP6 models: MPI-ESM-1-2-HAM and UKESM1-O-LL**

| Model | MPI-ESM-1-2-HAM | | UKESM1-O-LL | |
|---|---|---|---|---|
| Experiment ID | "hist-piAer" | "historical" | "hist-piAer" | "historical" |
| Activity ID | AerChemMIP | CMIP | AerChemMIP | CMIP |
| Institution ID | HAMMOZ-Consortium | HAMMOZ-Consortium | MOHC | MOHC |
| Nominal Resolution | 250 km | 250 km | 250 km | 250 km |
| Frequency | Monthly | Monthly | Monthly | Monthly |
| Variant label [data version] | r1i1p1f1 [20190627], r2i1p1f1 [20190627 /20190628][a], r3i1p1f1 [20190627 /20191218][b] | r1i1p1f1 [20190627], r2i1p1f1 [20190627], r3i1p1f1 [20190627 /20191218][b] | r1i1p1f2 [20190813], r2i1p1f2 [20191104], r3i1p1f2 [20200128] | r1i1p1f2 [20190406], r2i1p1f2 [20190502], r3i1p1f2 [20190502] |

[a]Data versions are 20190627 for cloud droplet number concentration (Nd), droplet effective radius (Re), liquid water path (LWP) and 20190628 for cloud cover (CC).
[b]Data versions are 20190627 for Nd, Re, LWP and 20191218 for CC.

summarised in Table 1. A brief description is provided below. Emissions from the Holuhraun eruption were specified in the VolcACI protocol[62], while other anthropogenic and natural emissions follow the AeroCom Phase III guidelines. Sea surface temperature and sea ice extent were prescribed using time-varying monthly mean values of the year 2014, as given in the AMIP data[66]. Model winds and surface pressure are nudged towards time-varying reanalysis data (e.g., ERA-Interim or MERRA2). The spin-up period was set to one year to allow the models to stabilise prior to the volcano simulations. Significant ACI signals and increase of cloud fraction have been detected using MODIS observations during September-October 2014[6]. Here, we focus on the month of October 2014, because the volcanic plume has been well dispersed over the whole North Atlantic by then, influencing a cloud regime spectrum that closely represents the global clouds[6,21]. Control and volcano scenarios were run with and without Holuhraun emissions for October 2014, respectively. The difference between the two scenarios dictates the clouds' response to volcanic aerosol perturbations.

### Cloud scheme and process sensitivities

ECHAM6.3-HAM2.3 has 47 vertical levels from the surface to the top level at 0.01 hPa and runs at T63 spectral horizontal resolution (1.875° × 1.875°). The two-moment cloud microphysics scheme (REF) is used as the default scheme for stratiform clouds in ECHAM-HAM[46]. This scheme simulates number concentrations and mass mixing ratios of cloud droplets and ice crystals prognostically. These hydrometeors interact with rain drops, snow, and water vapour, through various microphysical processes. A schematic figure showing the interconnections of microphysical processes in REF scheme can be found in ref. 67. Liquid cloud cover is diagnosed based on grid-mean RH and a height-dependent threshold RH[68]. To test different cloud microphysics and cloud cover schemes, we employed five different cloud scheme setups in ECHAM6.3-HAM2.3, including REF, REF-XR, P3, P3-XR, and PROG_CC. To test the sensitivity of individual processes, we perturbed specific microphysical processes in the REF and PROG_CC schemes to assess the sensitivity of liquid water path and cloud cover to these processes. Detailed parameter settings for the sensitivity studies are shown in Table 2.

### CMIP6 data for global cloud susceptibility

Global cloud property changes (i.e., cloud susceptibility) in response to present-day anthropogenic aerosol emissions are calculated through CMIP6 simulations and compared with Holuhraun-2014 eruption in this study to validate its global representativeness.

Among the six GCMs used in our study, only two models from CMIP6 provide complete outputs for both cloud microphysical and macro-physical properties: MPI-ESM-1-2-HAM and UKESM1-O-LL (equivalent to ECHAM6.3-HAM2.3 and UKESM1 in this study). We used paired "historical" and "hist-piAer" simulations (2000–2014) for MPI-ESM-1-2-HAM[69,70] and UKESM1-O-LL[71,72] from CMIP6. For each

experiment, three ensemble runs (r1, r2, r3) are available and used to assess the robustness of the results. The "historical" experiments simulate Earth's climate from 1850 to 2014, including both natural and anthropogenic forcings. In contrast, the "hist-piAer" experiments follow the same setup but exclude anthropogenic aerosols. Comparing these two experiments over the present-day period (2000–2014) allows us to isolate the influence of anthropogenic aerosols on global cloud properties. Consistent with Holuhraun-2014 simulations, monthly cloud variables (i.e. Nd, Re, LWP, CC) from "hist-piAer" and "historical" experiments are used to calculate cloud susceptibilities (i.e., $-d\ln R_e/d\ln N_d$, $d\ln LWP/d\ln N_d$, $d\ln COD/d\ln N_d$, and $d\ln LCC/d\ln N_d$). Note that liquid cloud cover is not available from the CMIP6 simulations, so total cloud cover is used to calculate cloud cover response. Detailed information of CMIP6 simulations is given in Table 3.

### Data availability

The GCM simulation data used to produce the figures are available at Zenodo (https://doi.org/10.5281/zenodo.16926288)[73]. The MODIS cloud products from Aqua (MYD08_L3) and Terra (MOD08_L3) are openly available from the Atmosphere Archive and Distribution System Distributed Active Archive Center of National Aeronautics and Space Administration (LAADS-DAAC, NASA) (https://ladsweb.modaps.eosdis.nasa.gov). CMIP6 data for MPI-ESM-1-2-HAM and UKESM1-O-LL models are openly available through the Earth System Grid Federation (ESGF) network (https://esgf-ui.ceda.ac.uk/cog/projects/esgf-ceda/).

### Code availability

The ECHAM-HAMMOZ model (ECHAM6.3-HAM2.3 and ECHAM6.3-SALSA2.0) is available to the scientific community under the HAMMOZ Software License Agreement at https://redmine.hammoz.ethz.ch/projects/hammoz/wiki/2_How_to_get_the_sources. The UKESM1 is released for use by UK researchers at https://ukesm.ac.uk/model-releases/. The CESM2 is available at: https://www.cesm.ucar.edu/models/cesm2/release_download.html. The documents and code of each module of CNRM-ESM2-1 are documented on the CNRM website (https://www.umr-cnrm.fr/cmip6). CAM5.3-Oslo (the atmospheric and aerosol module of NorESM) model code is available from GitHub: https://github.com/NorESMhub/.

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

## Acknowledgements

The ECHAM6.3-HAM2.3 model is developed by a consortium including ETH Zurich, Max-Planck-Institut für Meteorologie, Forschungszentrum Jülich, University of Oxford, the Finnish Meteorological Institute, and the Leibniz Institute for Tropospheric Research. The model simulations of the ECHAM6.3-HAM2.3 were supported by a grant from the Swiss National Supercomputing Centre (CSCS) under project ID s1144 and the Forth cluster at the University of Edinburgh. Y.W. acknowledges funding from the startup fund for lectureship from the University of Edinburgh. Y.W., D.N., and U.L. acknowledge financial support from Mr. Philippe Sarasin and the ETH Zürich Foundation. Y.C., Y.W., and G.M. acknowledge the support of the UKRI-NERC projects QUESTION (NE/B001024/1) and QR-CODE (NE/Z503800/1). D.P. would like to acknowledge support

from the NERC projects ADVANCE (NE/S015671/1) and CLOSURE (NE/W001713/1). G.J. acknowledges support from the European Union's Horizon 2020 CONSTRAIN grant (820829). We thank Prof. Jim Haywood (University of Exeter) and Dr Steffen Münch (PartnerRe, Switzerland) for useful discussions on main results. We acknowledge Dr. Brice Fourcart (previous at Météo-France–CNRS) for CNRM-ESM2-1 simulations and thank Dr. Inger Karset (previous at University of Oslo) and Prof. Trude Storelvmo (University of Oslo) for CAM5.3-Oslo simulations.

## Author contributions

Conceptualisation: Y.W., U.L., Y.C. Funding acquisition: Y.W., U.L., Y.C., G.M. Methodology and investigation: F.M., T.Y., G.J., D.P., P.F., M.M., P.N., A.L., Y.W., D.N., H.W., M.W., Y.C. Visualisation: Y.W. Writing – original draft: Y.W. Writing – review and editing: all co-authors.

## Competing interests

The authors declare no competing interests.
