## [Transparent Peer Review file · Nature Communications]

Challenges in global climate models to represent cloud response to aerosols: insights from volcanic eruptions

Corresponding Author: Dr Yu Wang

Version 0:

Reviewer comments:

Reviewer #1

(Remarks to the Author)

The paper investigates the ability of a selection of climate models to simulate the liquid cloud response to the massive injection of sulfur into the lower atmosphere associated with an effusive eruption in Iceland in 2014. The paper follows a similar methodology to reference 22, Malavelle et al. 2017. It is well organized and relatively concise in showing that climate models are fairly consistent in their errors in simulating this response. The models do relatively well on cloud optical depth responses, but underestimate the Twomey response and have almost no response in cloud fraction, compared to the satellite observations against which the models are tested. Trying many different parameterization schemes within one of the climate models, produces no significant change in these basic deficiencies.

Perhaps a little more detail on the ability of the satellite measurements to distinguish Re, LWP and LCC changes would be helpful. I imagine compensating errors in the measurements between Re, LWP and LCC, too.

The paper calls for a continued effort to understand these discrepancies and improve their simulation compared the observations, as this eruption and the associated data comprise a unique natural experiment. The paper does not comment on the special character of the observed situation relative to the global cloud problem. It occurs in high latitudes over the ocean in the summer season in the vicinity of Iceland. The sulfate injection is huge. It might be that fine tuning simulations to this particular situation, may not serve particularly well for the global problem. Also, I believe the cloud fractions in this location and season are very high, so that the logarithmic error seen here is not that significant in a real sense, and maybe not that robustly measured. That said, however, it well might be that the inability to simulate these data constitutes a fundamental inability of climate model parameterizations to simulate cloud fraction responses to condensation nuclei changes. I am not sure which it is.

Comments on text:

Line 111: 'theirs' does not make grammatical sense here.

137: Do you really want to say 'our previous study' here in relation to reference 22? I don't think the author list is identical. It also raises the question of what is really new here also.

192: When you say cloud albedo effect, to which of the feedbacks in Figure 1 are you referring. Cloud albedo is not shown there.

Reviewer #2

(Remarks to the Author)

The author provides an evaluation of aerosol-cloud interactions in six atmospheric models during the October 2014 Holuhraun volcanic event. The results are intriguing, and the manuscript is concise. However, I believe significant improvements in methodology and analysis are needed before publication. My detailed concerns are outlined below.

Major:

The analysis relies on a single month of model simulations, which may introduce significant noise due to the inherently chaotic nature of cloud dynamics and radiative effects, especially over short timescales. To strengthen robustness, I recommend quantifying uncertainty through ensemble simulations. If unavailable, multi-year control experiments could serve as a proxy to estimate noise magnitude.

The study's title and conclusions suggest broader relevance, but the connection between the Holuhraun volcanic event (a localized, sulfate-dominated aerosol source) and global aerosol-cloud interactions (ACI) remains unclear. If the models' responses are context-specific, consider revising the title to reflect this limitation (e.g., "Global climate models fail to reproduce cloud cover response to volcanic aerosol"). Clarifying the generalizability to other aerosol types or regions would strengthen the analysis. For example, a good start point is to show the vertical profile of the cloud change in these experiment (or a simple high and low cloud decomposition) and compare it with the historical aerosol only simulations.

Figure 1a shows CESM2 and CNRM-ESM with similar ACI values (if I understand correctly), yet Smith et al (2020) (<https://acp.copernicus.org/articles/20/9591/2020/acp-20-9591-2020.html#&gid=1&pid=1>) reports stronger ACI for CESM2 in CMIP6 (Table 6). This could be due to many reasons, include the two major concerns that I raise above. So, it is good to understand the reason as a sanity check for your results overall.

Reviewer #3

(Remarks to the Author)
See uploaded review.

[Editorial Note: See end of file]

Version 1:

Reviewer comments:

Reviewer #1

(Remarks to the Author)
Review of Challenges in global climate models to represent cloud response to aerosol: insights from volcanic eruptions. By Wang et al.

I have read over the responses to reviews, looked at some of the background literature and read through the revised paper. The main point of the paper is that models simulate the AOD change associated with the 2014 effusive eruption well, but do not get the apportionment between Re, LWP and CF correct, mainly because they do not predict a CF increase in response to the aerosols. The latter point is based on an acceptance of the Chen et al.(2022) paper, in which machine learning is used to predict these quantities as measured by MODIS, using meteorological data from ER5. I am not sure this result has been independently verified. Monthly mean data for October are used for the ML model, so about 19 Octobers in total, although I assume many 1-degree grid boxes. I am left feeling a bit skeptical about whether the Chen et al. (2022) residual estimates of the susceptibilities are robust enough to tune models to. I think this dependence should be a bit more emphasized in the paper. The paper makes it seem like the satellite data are rock solid, but there are major uncertainties in both the observational apportionment and its fitting with the ML method and these things are critical to the conclusions. It seems a bit worrisome that straightforward comparison of climatologies reveals no change in CF in 2014, but using the ML residual method introduces one.

Abstract and Conclusions: This work concludes that models underestimate cloud amount changes compared to MODIS, but previous work with MODIS data by McCoy and Hartmann(2015) Malavelle et al (2017) see large Twomey effect but little change in water path or fraction. Malavelle et al(2017) use MODIS data and state that "cloud amount changes are undetectable" What changed in the interim to make cloud amount changes a significant effect? This seems to be based on the work of Wang et al. (2022) that use ML modeling of October monthly mean data to estimate the meteorological effects on cloud properties. Although straightforward average comparisons suggest no change in LWP or CF between September or October of 2014 with the long-term climatology for those months, the ML method finds an increase in cloud fraction in October 2014 in response to aerosols. This seems to suggest that the ML model predicted a decrease in CF in October 2014 that was not observed and then attributed the difference to the aerosols. I have not studied the methods thoroughly, but I am somewhat unconvinced, particularly since the September data were not used, which would have given an independent data point. I find the simple result that no significant difference in cloud fraction was observed in 2014 more compelling than the residual of a ML computation based on monthly data (20 months total) and the residual derived therefrom. What am I missing?

Abstract concludes with " , and support Net-Zero Policymaking" makes it seem like the purpose of scientific research is to support a particular policy option. I strongly recommend against using this phrasing.

45-46 "none of state-of-the-art climate models are able to fully account for this recent warming"

This statement is hyperbolic and makes it seem like we don't know what we are doing, and it also is unbalanced in a similar way to people who said warming had stopped after 1998. There was a big jump in 2023 from 2022, but equally remarkable is the small amount of warming from 1998 to 2014. Smoothing over these anomalies, the overall trend has continued apace at about 0.2K/decade, which is what the models are predicting (if this trend is extrapolated from 2024 to 2100 it reaches about 3K anomaly in 2100, in good agreement with central estimates for a scenario that does not rapidly reduce CO₂ emissions.). One should not expect models to exactly explain one year's anomaly because of the major influences of natural variability and the uneven, sometimes stepwise response to human influences. The statement that models cannot "fully account" for the anomaly in 2023 is probably strictly true for every year in the record. It is not helpful to make things seem more uncertain than they probably are.

82 "both positive and negative results reported" Also a lot of zero effect on LWP and CF. Can also depend on meteorology and the magnitude of the aerosol source compared to background. Evidence that background levels decrease the effect of emissions of sulfate and the effect is also not linear, diminishing for very large sulfate loading. Therefore one case like Iceland cannot be the silver bullet for the global problem.

95-96 Are there references for this statement that the observational uncertainty has been removed? Also, explain why the observational response of CF is now large, compared to earlier observational work that suggested it is small. I would be skeptical if it is based on AI methodologies that do not have a strong physical basis and explanation. The LWC and CF changes do now show up in simple averages, only the Twomey effect.

101 – "we suggest that the wider climate community takes the Holuhraun-2014 eruption as an anchor case" Several papers have already investigated the eruptions effect on clouds and compared with models, so what is being suggested is not new. Malavelle et al. (2017) Christensen et al. (2022), Marshall et al.(2022), Chen et al. (2022), Jordan et al.(2024), Feingold et al. (2024), Chen et al.(2024, Hawaiian Volcanoes), Peace et al.(2024), Zoega et al.(2025a), Zoega et al.(2025b), Marelle et al. (2025).

156-157: Getting the albedo change right is central. It is also much more accurately measured that apportionment of albedo change to Re, LWP and CF. I wonder if it can be shown that the observational apportionment between Re, LWP and CF is less uncertain than that in the models. Based on my reading of the Chen et al. (2022) paper from which the ML estimates come, I am not sure, Is the ML method used here good enough to use to change models?

References:

- Anoruo, C. M., O. C. Ibe and K. N. Ndubuisi (2023). "Aerosol Load-Cloud Cover Correlation: A Potential Clue for the Investigation of Aerosol Indirect Impact on Climate of Europe and Africa." *Aerosol Science and Engineering* 7(1): 23-35.
- Bellouin, N., J. Quaas, E. Gryspeerdt, S. Kinne, P. Stier, D. Watson-Parris, O. Boucher, K. S. Carslaw, M. Christensen, A. L. Daniau, J. L. Dufresne, G. Feingold, S. Fiedler, P. Forster, A. Gettelman, J. M. Haywood, U. Lohmann, F. Malavelle, T. Mauritsen, D. T. McCoy, G. Myhre, J. Mülmenstädt, D. Neubauer, A. Possner, M. Rugenstein, Y. Sato, M. Schulz, S. E. Schwartz, O. Sourdeval, T. Storelvmo, V. Toll, D. Winker and B. Stevens (2020). "Bounding Global Aerosol Radiative Forcing of Climate Change." *Reviews of Geophysics* 58(1).
- Bender, F. A. M., V. Jung, A. Staffansdotter, T. Lord and S. Undorf (2024). "Machine Learning Approach to Investigating the Relative Importance of Meteorological and Aerosol-Related Parameters in Determining Cloud Microphysical Properties." *Tellus Series B-Chemical and Physical Meteorology* 76(1): 1-18.
- Chen, Y., J. Haywood, Y. Wang, F. Malavelle, G. Jordan, A. Peace, D. G. Partridge, N. Cho, L. Oreopoulos, D. Grosvenor, P. Field, R. P. Allan and U. Lohmann (2024). "Substantial cooling effect from aerosol-induced increase in tropical marine cloud cover." *Nature Geoscience* 17(5): 404-+.
- Chen, Y., J. Haywood, Y. Wang, F. Malavelle, G. Jordan, D. Partridge, J. Fieldsend, J. De Leeuw, A. Schmidt, N. Cho, L. Oreopoulos, S. Platnick, D. Grosvenor, P. Field and U. Lohmann (2022). "Machine learning reveals climate forcing from aerosols is dominated by increased cloud cover." *Nature Geoscience* 15(8): 609-+.
- Christensen, M. W., A. Gettelman, J. Cermak, G. Dagan, M. Diamond, A. Douglas, G. Feingold, F. Glassmeier, T. Goren, D. P. Grosvenor, E. Gryspeerdt, R. Kahn, Z. Q. Li, P. L. Ma, F. Malavelle, I. L. McCoy, D. T. McCoy, G. McFarquhar, J. Mülmenstädt, S. Pal, A. Possner, A. Povey, J. Quaas, D. Rosenfeld, A. Schmidt, R. Schrödner, A. Sorooshian, P. Stier, V. Toll, D. Watson-Parris, R. Wood, M. X. Yang and T. L. Yuan (2022). "Opportunistic experiments to constrain aerosol effective radiative forcing." *Atmospheric Chemistry and Physics* 22(1): 641-674.
- Diamond, M. S., H. M. Director, R. Eastman, A. Possner and R. Wood (2020). "Substantial Cloud Brightening From Shipping in Subtropical Low Clouds." *Agu Advances* 1(1).
- Feingold, G., V. P. Ghate, L. M. Russell, P. Blosser, W. Cantrell, M. W. Christensen, M. S. Diamond, A. Gettelman, F. Glassmeier, E. Gryspeerdt, J. Haywood, F. Hoffmann, C. M. Kaul, M. Lebsack, A. C. McComiskey, D. T. McCoy, Y. Ming, J. Muelmenstaedt, A. Possner, P. Prabhakaran, P. K. Quinn, K. S. Schmidt, R. A. Shaw, C. E. Singer, A. Sorooshian, V. Toll, J. S. Wan, R. Wood, F. Yang, J. H. Zhang and X. Zheng (2024). "Physical science research needed to evaluate the viability and risks of marine cloud brightening." *Science Advances* 10(12).
- Goren, T., O. Sourdeval, J. Kretschmar and J. Quaas (2023). "Spatial Aggregation of Satellite Observations Leads to an Overestimation of the Radiative Forcing Due To Aerosol-Cloud Interactions." *Geophysical Research Letters* 50(18).
- Haywood, J. M., O. Boucher, C. Lennard, T. Storelvmo, S. Tilmes and D. Vioni (2025). "World Climate Research Programme lighthouse activity: an assessment of major research gaps in solar radiation modification research." *Frontiers in Climate* 7.
- Jordan, G., F. Malavelle, Y. Chen, A. Peace, E. Duncan, D. G. Partridge, P. Kim, D. Watson-Parris, T. Takemura, D. Neubauer, G. Myhre, R. Skeie, A. Laakso and J. Haywood (2024). "How well are aerosol-cloud interactions represented in

climate models? - Part 1: Understanding the sulfate aerosol production from the 2014-15 Holuhraun eruption." *Atmospheric Chemistry and Physics* 24(3): 1939-1960.

Malavelle, F. F., J. M. Haywood, A. Jones, A. Gettelman, L. C. Lariße, S. Bauduin, R. P. Allan, I. H. H. Karset, J. E. Kristjánsson, L. Oreopoulos, N. Y. C. Ho, D. Lee, N. Bellouin, O. Boucher, D. P. Grosvenor, K. S. Carslaw, S. Dhomse, G. W. Mann, A. Schmidt, H. Coe, M. E. Hartley, M. Dalvi, A. A. Hill, B. T. Johnson, C. E. Johnson, J. R. Knight, F. M. O'Connor, D. G. Partridge, P. Stier, G. Myhre, S. Platnick, G. L. Stephens, H. Takahashi and T. Thordarson (2017). "Strong constraints on aerosol-cloud interactions from volcanic eruptions." *Nature* 546(7659): 485-491.

Marelle, L., G. Myhre, J. L. Thomas and J. C. Raut (2025). "Aerosol Background Concentrations Influence Aerosol-Cloud Interactions as Much as the Choice of Aerosol-Cloud Parameterization." *Geophysical Research Letters* 52(8).

Marshall, L. R., E. C. Maters, A. Schmidt, C. Timmreck, A. Robock and M. Toohey (2022). "Volcanic effects on climate: recent advances and future avenues." *Bulletin of Volcanology* 84(5).

Peace, A. H., Y. Chen, G. Jordan, D. G. Partridge, F. Malavelle, E. Duncan and J. M. Haywood (2024). "In-plume and out-of-plume analysis of aerosol-cloud interactions derived from the 2014-2015 Holuhraun volcanic eruption." *Atmospheric Chemistry and Physics* 24(16): 9533-9553.

Rosenfeld, D., A. Kokhanovsky, T. Goren, E. Gryspeerd, O. Hasekamp, H. L. Jia, A. Lopatin, J. Quaas, Z. X. Pan and O. Sourdeval (2023). "Frontiers in Satellite-Based Estimates of Cloud-Mediated Aerosol Forcing." *Reviews of Geophysics* 61(4).

Toll, V. (2022). "Polluted skies are cloudier." *Nature Geoscience* 15(8): 601-602.

Wang, X., F. Y. Mao, D. Rosenfeld, Y. N. Zhu, Z. X. Pan, Y. Cao, L. Zang, X. Lu and W. Gong (2025). "Volcanic aerosols lend causality to the indicated substantial susceptibility of clouds to aerosol over global oceans." *Npj Climate and Atmospheric Science* 8(1).

Wang, X., F. Y. Mao, Y. N. Zhu, D. Rosenfeld, Z. X. Pan, L. Zang, X. Lu, F. Liu and W. Gong (2024). "Hidden Large Aerosol-Driven Cloud Cover Effect Over High-Latitude Ocean." *Journal of Geophysical Research-Atmospheres* 129(13).

Zoëga, T., T. Storelvmo and K. Krüger (2025a). "Arctic warming from a high-latitude effusive volcanic eruption." *Scientific Reports* 15(1).

Zoëga, T., T. Storelvmo and K. Krüger (2025b). "Modelled surface climate response to effusive Icelandic volcanic eruptions: sensitivity to season and size." *Atmospheric Chemistry and Physics* 25(5): 2989-3010.

Reviewer #2

(Remarks to the Author)

My comments have been addressed appropriately, and I recommend acceptance of the manuscript.

Version 2:

Reviewer comments:

Reviewer #1

(Remarks to the Author)

I appreciate the energy with which the authors have addressed my comments on the previous version of this paper. I think my concerns have been adequately addressed and I now recommend publication.

**Response to Reviewer #1 (manuscript NCOMMS-25-29057):**

**General comments:**

The paper investigates the ability of a selection of climate models to simulate the liquid cloud
response to the massive injection of sulfur into the lower atmosphere associated with an
effusive eruption in Iceland in 2014. The paper follows a similar methodology to reference 22,
Malavelle et al. 2017. It is well organized and relatively concise in showing that climate models
are fairly consistent in their errors in simulating this response. The models do relatively well
on cloud optical depth responses, but underestimate the Twomey response and have almost no
response in cloud fraction, compared to the satellite observations against which the models are
tested. Trying many different parameterization schemes within one of the climate models,
produces no significant change in these basic deficiencies.

**Response:**

We thank the reviewer for the positive feedback and constructive comments. All suggestions
have been carefully considered, and the manuscript has been revised accordingly. In particular,
we have addressed the global representativeness of the Holuhraun-2014 eruption by comparing
it with global cloud responses to anthropogenic aerosol emissions in the present-day period
using CMIP6 outcomes. A new section in the *Results* has been added to discuss the global
representativeness. Please find our detailed point-by-point responses below. A change-tracked
revision of the manuscript has also been uploaded.

**Point-by-point response to specific comments:**

1) Perhaps a little more detail on the ability of the satellite measurements to distinguish Re,
LWP and LCC changes would be helpful. I imagine compensating errors in the measurements
between Re, LWP and LCC, too.

**Response:**

Cloud properties such as droplets effective radius (Re), liquid water path (LWP), liquid cloud
cover (LCC) from the MODIS Terra & Aqua satellites provide a unique dataset of global and
long-term consistent observations of aerosols and clouds¹, representing one of the most widely
used datasets for aerosol-cloud interaction studies (e.g., refs^{2, 3, 4, 5, 6, 7} and references therein).
However, we agree with the reviewer that uncertainties in MODIS-retrievals do exist and may
propagate to the analyses of aerosol-cloud interactions (ACI).

Nevertheless, we believe our analysis and the conclusion are not largely impacted by satellite
uncertainty. Because, Chen et al. (2022)⁷ demonstrated that uncertainties in MODIS retrievals
can be effectively minimised in the derived ACI signals used ML-surrogate of MODIS:
“*Uncertainty in the MODIS retrievals can be decomposed into systematic errors and random*

errors. Random errors are greatly suppressed by averaging over a geographical region of
 thousands of kilometres⁸, while systematic errors are largely cancelled when taking differences
 between MODIS and ML-MODIS².”

Furthermore, we believe the cloud susceptibility signal detected by Chen et al. (2022)⁷, which
 is used in this study, is indeed due to aerosol perturbation rather than bias from MODIS errors.
 Because Chen et al. (2024)⁶ applied the same machine-learning approach to the trade wind
 regions of Hawaii-volcanoes; they find no ACI signal in the upwind clean region but a strong
 ACI signal only in the downwind region polluted by volcanic aerosols (see Fig. R1). If the ACI
 signal was due to satellite error, we should expect similar signals for the downstream and
 upstream regions.

**Figure R1. Cloud susceptibility to volcanic aerosol:** (a) downwind polluted region; (b)
 upwind clean region. Source: figure is adapted from Figs. 2 and S2 in Chen et al. (2024)⁶

We have added words in the introduction to emphasise this key point when introducing the
 satellite-derived observational constraints⁷.

“The challenge of confounding meteorology has been addressed in a recent work⁷, using machine-learning-based counterfactual clouds without volcanic influence to contrast real-world observations (i.e. clouds with volcanic influence). The detected ACI signals effectively minimise the noise from satellite observational uncertainty, because random

errors in observations are mostly cancelled out by averaging across an extensive region spanning millions of square kilometres, and systematic errors are largely suppressed when comparing differences between satellite and a machine-learned surrogate satellite⁷. This methodology has enabled robust estimation of cloud susceptibility at a large scale from satellite observations, making it possible to calibrate cloud susceptibility in GCMs.”

2) The paper calls for a continued effort to understand these discrepancies and improve their
simulation compared the observations, as this eruption and the associated data comprise a
unique natural experiment. The paper does not comment on the special character of the
observed situation relative to the global cloud problem. It occurs in high latitudes over the
ocean in the summer season in the vicinity of Iceland. The sulfate injection is huge. It might be
that fine tuning simulations to this particular situation, may not serve particularly well for the
global problem.

**Response:**

Great point. We believe this concern has been addressed by our further analysis of CMIP6
simulations to demonstrate the global representativeness of ACI in Holuhraun-2014 eruption.
In addition, as suggested by Reviewer-2, we have also performed ensemble analysis to
demonstrate the robustness of our analysis. Detailed discussion has been added in a new section
in the *Results* part of the main text, see below:

A new section in the *Result*:

“Global representativeness and robustness of the Holuhraun eruption

To evaluate the global representativeness of the Holuhraun-2014 eruption case, we compared its cloud regime distribution and cloud susceptibilities to global ones. Malavelle et al.³ and Chen et al.⁷ have shown that the Holuhraun-2014 eruption occurred in a region encompassing the full range of cloud regimes, whose distribution is closely matched with the global distribution observed by satellites from 2002 to 2014 (see Extended Data Fig.1 in Chen et al.⁷, copied to below). In addition, we assessed cloud susceptibilities at a global scale resulting from anthropogenic aerosol emissions throughout 2000 to 2014 in CMIP6 simulations (MPI-ESM-1-2-HAM and UKESM1-0-LL, equivalent to ECHAM6.3-HAM2.3 and UKESM1 in this study, respectively). We compare CMIP6 simulations of “historical” and “hist-piAer” experiments, representing with and without anthropogenic aerosol influence respectively; details are given in the Methods. The susceptibilities of clouds to anthropogenic aerosol emissions at the global scale are consistent with the Holuhraun-2014 eruption in our GCMs simulations (Fig. 1a). The consistency in both cloud regimes distribution and cloud susceptibilities suggests that the Holuhraun-2014 eruption provides a valid and representative analogue for understanding global cloud responses to aerosol perturbations due to human emissions in recent years.

To strengthen robustness of our results, we quantified model uncertainty caused by the inherently chaotic nature of cloud dynamics and radiative effects through ensemble simulations. We used ECHAM6.3-HAM2.3 as a showcase model and explored uncertainty across five representative experiments, each reflecting distinct cloud schemes and process sensitivities, with ten ensemble simulations per experiment. We find only a small degree of uncertainty is introduced by model internal variability. For cloud cover the ensemble uncertainty is a magnitude smaller than the gap between GCMs and the observation (Fig.2). Furthermore, we demonstrate the robustness in global representativeness using ensembles in CMIP6 simulations (three ensembles for each). Again, a negligible difference is found between CMIP6 ensembles (Fig.2). These well demonstrate the robustness of the key conclusions and its global representativeness in this study.

[REDACTED]

**Extended Data Fig.1 in Chen et al. (2022)⁷. [REDACTED]**

We have also added a new section in the *Methods* part to introduce the details of CMIP6

experiments that were used in this study, as shown below:

A new section in the *Methods* to describe the CMIP6 simulations used in this study:

“CMIP6 data for global cloud susceptibility

Global cloud property changes (i.e., cloud susceptibility) in response to present-day anthropogenic aerosol emissions are calculated through CMIP6 simulations and compared with Holuhraun-2014 eruption in this study to validate its global representativeness.

Among the six GCMs used in our study, only two models from CMIP6 provide complete outputs for both cloud microphysical and macro-physical properties: MPI-ESM-1-2-HAM and UKESM1-0-LL (equivalent to ECHAM6.3-HAM2.3 and UKESM1 in this study). We used paired “historical” and “hist-piAer” simulations (2000-2014) for MPI-ESM-1-2-HAM^{10, 11} and UKESM1-0-LL^{12, 13} from CMIP6. For each experiment, three ensemble runs (r1, r2, r3) are available and used to assess the robustness of the results. The “historical” experiment simulates Earth’s climate from 1850 to 2014, including both natural and anthropogenic forcings. In contrast, the “hist-piAer” experiment follows the same setup but excludes anthropogenic aerosols. Comparing these two experiments over the present-day period (2000-2014) allows us to isolate the influence of anthropogenic aerosols on global cloud properties. Consistent with Holuhraun-2014 simulations, monthly cloud variables (i.e. Nd, Re, LWP, CC) from “hist-piAer” and “historical” experiments are used to calculate cloud susceptibilities (i.e., $-\text{dln}R_e/\text{dln}N_a$, $\text{dln}LWP/\text{dln}N_a$, $\text{dln}COD/\text{dln}N_a$, and $\text{dln}LCC/\text{dln}N_a$). Note that liquid cloud cover is not available from the CMIP6 simulations, so total cloud cover is used to calculate cloud cover response. Detailed information of CMIP6 simulations is given in Table 3.”

Table 3 (newly added). Experiment information for selected CMIP6 models: MPI-ESM-1-2-HAM and UKESM1-0-LL.

Model	MPI-ESM-1-2-HAM		UKESM1-0-LL	
Experiment ID	“hist-piAer”	“historical”	“hist-piAer”	“historical”
Activity ID	AerChemMIP	CMIP	AerChemMIP	CMIP
Institution ID	HAMMOZ- Consortium	HAMMOZ- Consortium	MOHC	MOHC
Nominal Resolution	250 km	250 km	250 km	250 km
Frequency	Monthly	Monthly	Monthly	Monthly
Variant label	r1i1p1f1	r1i1p1f1	r1i1p1f2	r1i1p1f2
[data version]	[20190627],	[20190627], r2i1p1f1 [20190627],	[20190813], r2i1p1f2 [20191104],	[20190406], r2i1p1f2 [20190502],

r2ilplf1	r3ilplf1	r3ilplf2	r3ilplf2
[20190627	[20190627	[20200128]	[20190502]
/20190628]#,	/20191218]*		
r3ilplf1			
[20190627			
/20191218]*			

#Note: data versions are 20190627 for Nd, Re, LWP and 20190628 for CC.

**Note: data versions are 20190627 for Nd, Re, LWP and 20191218 for CC.*

3) Also, I believe the cloud fractions in this location and season are very high, so that the
 logarithmic error seen here is not that significant in a real sense, and maybe not that robustly
 measured. That said, however, it well might be that the inability to simulate these data
 constitutes a fundamental inability of climate model parameterizations to simulate cloud
 fraction responses to condensation nuclei changes. I am not sure which it is.

**Response:**

The reviewer is right that the liquid cloud fractions in this location and specific season (41%)
 are higher than the multi-year global annual averages (30% over ocean) from 2000 to 2020
 (excluding 2014). We chose and still prefer to use the logarithmic form in our analysis because
 it is an appropriate and better way to evaluate cloud susceptibility to aerosol perturbation, for
 the following reasons:

1) The logarithmic form implies the relative change, for example:

$$d \ln N_d = \ln N_{d1} - \ln N_{d0} = \ln\left(\frac{N_{d1}}{N_{d0}}\right)$$

The bracket term shows that the logarithmic change is equivalent to relative change.

Global climate models are tuned to match the observations of global temperature or
 TOA radiation flux. But each model's cloud properties, including the baseline
 (preindustrial) N_d , can vary widely (a 33-63 $\#/cm^3$ range across nine global climate
 models averaged over the global ocean areas¹⁴). However, if all models were to follow
 a similar aerosol-cloud interaction principle (or similar cloud susceptibility, e.g.:
 $d\ln LCC/d\ln N_d$), then the relative changes of cloud optical depth and cloud cover (and
 hence the reflected sunlight) will be proportional to the relative changes of N_d , but
 relatively insensitive to the very uncertain baseline.

2) Given the aforementioned importance of relative change, the logarithmic form (hence
relative changes) is commonly used in many previous modelling and observational
aerosol-cloud-climate studies. For example, Ghan et al. (2016)¹⁴ compared the
performance of nine global climate models using the logarithmic form; Rosenfeld et al.
(2019)¹⁵ and Toll et al. (2019)² used observationally-derived logarithmic terms to
discuss aerosol-cloud interactions.

Therefore, we performed our analysis in the logarithmic form for better constraining models,
more readily for comparison with other studies, and easier to be directly used/compared by the
community in future studies.

In summary, given the reasons provided in the Points 1-3, we have well demonstrated that the
ACI signal (cloud susceptibility) is clearly detected from the satellite observations, and that our
analysis points towards a fundamental inability of climate model parameterisations to simulate
cloud cover responses to aerosol changes, and this is a critical common gap with global
representativeness rather than a context-specified issue.

**Comments on text:**

Line 111: ‘theirs’ does not make grammatical sense here.

**Response:**

Thanks for carefulness, corrected.

137: Do you really want to say ‘our previous study’ here in relation to reference 22? I don’t
think the author list is identical. It also raises the question of what is really new here also.

**Response:**

Corrected to “*a previous study*”. Although Malavelle and some of the co-authors are also
involved in this study, we agree that the author list is not identical between Malavelle et al.
(2017)³ and this study.

192: When you say cloud albedo effect, to which of the feedbacks in Figure 1 are you referring.
Cloud albedo is not shown there.

**Response:**

Corrected to “*cloud optical depth response*”. Cloud optical depth (COD) has usually been used
as an indicator of cloud albedo; however, we agree with the reviewer that they are not the same
and albedo is not directly shown in Figure 1. We therefore have corrected it to avoid the
confusion.

**References:**

- 1. Platnick S, *et al.* The MODIS Cloud Optical and Microphysical Products: Collection 6
Updates and Examples From Terra and Aqua. *IEEE Transactions on Geoscience and*
*Remote Sensing* **55**, 502-525 (2017).
- 2. Toll V, Christensen M, Quaas J, Bellouin N. Weak average liquid-cloud-water response
to anthropogenic aerosols. *Nature* **572**, 51-55 (2019).
- 3. Malavelle FF, *et al.* Strong constraints on aerosol–cloud interactions from volcanic
eruptions. *Nature* **546**, 485-491 (2017).
- 4. Quaas J, *et al.* Robust evidence for reversal of the trend in aerosol effective climate
forcing. *Atmos Chem Phys* **22**, 12221-12239 (2022).
- 5. Yuan T, *et al.* Observational evidence of strong forcing from aerosol effect on low cloud
coverage. *Science Advances* **9**, eadh7716 (2023).
- 6. Chen Y, *et al.* Substantial cooling effect from aerosol-induced increase in tropical
marine cloud cover. *Nature Geoscience* **17**, 404-410 (2024).
- 7. Chen Y, *et al.* Machine learning reveals climate forcing from aerosols is dominated by
increased cloud cover. *Nature Geoscience* **15**, 609-614 (2022).
- 8. Grosvenor DP, *et al.* Remote Sensing of Droplet Number Concentration in Warm
Clouds: A Review of the Current State of Knowledge and Perspectives. *Reviews of*
*Geophysics* **56**, 409-453 (2018).
- 9. Oreopoulos L, Cho N, Lee D, Kato S. Radiative effects of global MODIS cloud regimes.
*Journal of Geophysical Research: Atmospheres* **121**, 2299-2317 (2016).
- 10. Neubauer D, *et al.* HAMMOZ-Consortium MPI-ESM1.2-HAM model output prepared
for CMIP6 CMIP historical. Earth System Grid Federation.
<https://doi.org/10.22033/ESGF/CMIP6.5016>. (2019).
- 11. Neubauer D, *et al.* HAMMOZ-Consortium MPI-ESM1.2-HAM model output prepared
for CMIP6 AerChemMIP hist-piAer. Earth System Grid Federation.
<https://doi.org/10.22033/ESGF/CMIP6.5007>. (2019).
- 12. Tang Y, *et al.* MOHC UKESM1.0-LL model output prepared for CMIP6 CMIP
historical. Earth System Grid Federation. <https://doi.org/10.22033/ESGF/CMIP6.6113>.
(2019).
- 13. O'Connor F. MOHC UKESM1.0-LL model output prepared for CMIP6 AerChemMIP
hist-piAer. Earth System Grid Federation. <https://doi.org/10.22033/ESGF/CMIP6.6062>.
(2019).
- 14. Ghan S, *et al.* Challenges in constraining anthropogenic aerosol effects on cloud
radiative forcing using present-day spatiotemporal variability. *Proceedings of the*
*National Academy of Sciences* **113**, 5804-5811 (2016).

- 15. Rosenfeld D, Zhu Y, Wang M, Zheng Y, Goren T, Yu S. Aerosol-driven droplet
concentrations dominate coverage and water of oceanic low-level clouds. *Science* **363**,
eaav0566 (2019).

**Response to Reviewer #2 (manuscript NCOMMS-25-29057):**

**General comments:**

The author provides an evaluation of aerosol-cloud interactions in six atmospheric models
during the October 2014 Holuhraun volcanic event. The results are intriguing, and the
manuscript is concise. However, I believe significant improvements in methodology and
analysis are needed before publication. My detailed concerns are outlined below.

**Response:**

We thank the reviewer for the positive feedback and constructive comments. All suggestions
have been carefully considered, and the manuscript has been revised accordingly. In particular,
we have addressed the global representativeness of the Holuhraun-2014 eruption by comparing
it with global cloud responses to anthropogenic aerosol emissions in the present-day period
using CMIP6 data. Model uncertainty has been evaluated through ensemble runs in a showcase
model, ECHAM6.3-HAM2.3 and CMIP6. A new section discussing this comparison has been
added to the *Results*. Please find our detailed point-by-point responses below. A change-tracked
revision of the manuscript has also been uploaded.

**Point-by-point responses to specific comments:**

1) The analysis relies on a single month of model simulations, which may introduce significant
noise due to the inherently chaotic nature of cloud dynamics and radiative effects, especially
over short timescales. To strengthen robustness, I recommend quantifying uncertainty through
ensemble simulations. If unavailable, multi-year control experiments could serve as a proxy to
estimate noise magnitude.

**Response:**

We agree with the reviewer that ensemble simulations of Holuhraun-2014 eruption are helpful
to quantify uncertainty of model internal variability and thus strengthen robustness of results.
Although ensemble runs are not available for multi-model simulations from the AeroCom
VolACI project, we used ECHAM6.3-HAM2.3 as a showcase to quantify this uncertainty. To
better represent the uncertainty in diverse modelling configurations, we performed
ECHAM6.3-HAM2.3 simulations with 5 distinct configurations of cloud schemes and relevant
processes (each with 10 ensembles). We found a minor uncertainty in all ECHAM6.3-HAM2.3
and CMIP6 ensembles. Detailed discussion has been added in a new section (“Global
representativeness and robustness of the Holuhraun eruption”) in the *Results* of main text, as
shown following point-2.

2) The study’s title and conclusions suggest broader relevance, but the connection between the
Holuhraun volcanic event (a localized, sulfate-dominated aerosol source) and global aerosol-
cloud interactions (ACI) remains unclear. If the models’ responses are context-specific,
consider revising the title to reflect this limitation (e.g., “Global climate models fail to
reproduce cloud cover response to volcanic aerosol”). Clarifying the generalizability to other
aerosol types or regions would strengthen the analysis. For example, a good start point is to
show the vertical profile of the cloud change in these experiment (or a simple high and low
cloud decomposition) and compare it with the historical aerosol only simulations.

**Response:**

Thanks for the suggestion. As demonstrated in the further analysis with CMIP6 as shown below
in the new section: “Global representativeness and robustness of the Holuhraun eruption”, the
Holuhraun-2014 eruption shows a validated and robust global representativeness of ACI.
Reflecting on the suggestions on the title from Reviewers-2&3, we have changed to a new title:
“*Challenges in global climate models to represent cloud response to aerosol: insights from*
*volcanic eruptions*”.

Regarding the generalisation of aerosol types, we focused on cloud susceptibilities in this study,
cloud properties (Re, LWP, LCC) changes as a response to an increase in Nd. These cloud
adjustments happened after activation of aerosol particles to serve as cloud condensation nuclei
(CCN). The CCN can come from any aerosol type, and is not specific to volcanic aerosol.
Therefore, aerosol types are not sensitive in this study as long as they are activated and provided
increased Nd.

A new section in the *Results*:

“Global representativeness and robustness of the Holuhraun eruption

To evaluate the global representativeness of the Holuhraun-2014 eruption case, we compared its cloud regime distribution and cloud susceptibilities to global ones. Malavelle et al.¹ and Chen et al.² have shown that the Holuhraun-2014 eruption occurred in a region encompassing the full range of cloud regimes, whose distribution is closely matched with the global distribution observed by satellites from 2002 to 2014 (see Extended Data Fig.1 in Chen et al.², copied to below). In addition, we assessed cloud susceptibilities at a global scale resulting from anthropogenic aerosol emissions throughout 2000 to 2014 in CMIP6 simulations (MPI-ESM-1-2-HAM and UKESM1-0-LL, equivalent to ECHAM6.3-HAM2.3 and UKESM1 in this study, respectively). We compare CMIP6 simulations of “historical” and “hist-piAer” experiments, representing with and without anthropogenic aerosol influence respectively; details are given in the Methods. The susceptibilities of clouds to anthropogenic aerosol emissions at the global scale are consistent with the Holuhraun-2014 eruption in our GCMs simulations (Fig. 1a). The consistency in both cloud regimes distribution and cloud susceptibilities suggests that the Holuhraun-2014 eruption provides a valid and representative analogue for understanding global cloud responses to aerosol perturbations due to human emissions in recent years.

To strengthen robustness of our results, we quantified model uncertainty caused by the inherently chaotic nature of cloud dynamics and radiative effects through ensemble simulations. We used ECHAM6.3-HAM2.3 as a showcase model and explored uncertainty across five representative experiments, each reflecting distinct cloud schemes and process sensitivities, with ten ensemble simulations per experiment. We find only a small degree of uncertainty is introduced by model internal variability. For cloud cover the ensemble uncertainty is a magnitude smaller than the gap between GCMs and the observation (Fig.2). Furthermore, we demonstrate the robustness in global representativeness using ensembles in CMIP6 simulations (three ensembles for each). Again, a negligible difference is found between CMIP6 ensembles (Fig.2). These well demonstrate the robustness of the key conclusions and its global representativeness in this study.”

Figure 1a (updated for CMIP6). Intercomparison of liquid cloud properties response to aerosol perturbations (i.e., cloud susceptibilities) from volcanic degassing at Holuhraun, Iceland (October 2014), based on multiple model simulations with varying cloud schemes and process sensitivities, alongside observational constraints². **a** Six GCMs from this study, representative of the Holuhraun region behaviour, and two GCMs from the Coupled Model Intercomparison Project Phase 6 (CMIP6), representative of the global behaviour.

Figure 2 (newly added). Intercomparison of liquid cloud properties response to aerosol perturbations (i.e., cloud susceptibilities) from Holuhraun-2014 eruption in ECHAM6.3-HAM2.3 with different cloud schemes and sensitivities (each experiment with ten ensemble runs, ensemble mean values and one standard deviation are shown) and CMIP6 data (MPI-ESM1-2-HAM and UKESM1-0-LL, each with three ensemble runs). The violin plots represent 90% confidence intervals of machine-learning-based observational constraints², with the inner box showing the 25th, 50th, and 75th percentiles. Note: the three ensemble runs for each CMIP6 model are all plotted, but they overlap with each other due to negligible differences.

[REDACTED]

**Extended Data Fig.1 in Chen et al. (2022)². [REDACTED]**

73

74 We have also added a new section in the *Methods* part to introduce the details of CMIP6
75 experiments that were used in this study, as shown below:

A new section in the *Methods* to describe the CMIP6 simulations used in this study:

“CMIP6 data for global cloud susceptibility

Global cloud property changes (i.e., cloud susceptibility) in response to present-day anthropogenic aerosol emissions are calculated through CMIP6 simulations and compared with Holuhraun-2014 eruption in this study to validate its global representativeness.

Among the six GCMs used in our study, only two models from CMIP6 provide complete outputs for both cloud microphysical and macro-physical properties: MPI-ESM-1-2-HAM and UKESM1-0-LL (equivalent to ECHAM6.3-HAM2.3 and UKESM1 in this study). We used paired “historical” and “hist-piAer” simulations (2000-2014) for MPI-ESM-1-2-HAM^{4, 5} and UKESM1-0-LL^{6, 7} from CMIP6. For each experiment, three ensemble runs (r1, r2, r3) are available and used to assess the robustness of the results. The “historical” experiment simulates Earth’s climate from 1850 to 2014, including both natural and anthropogenic forcings. In contrast, the “hist-piAer” experiment follows the same setup but excludes anthropogenic aerosols. Comparing these two experiments over the present-day period (2000-2014) allows us to isolate the influence of anthropogenic aerosols on global cloud properties. Consistent with Holuhraun-2014 simulations, monthly cloud variables (i.e. Nd, Re, LWP, CC) from “hist-piAer” and “historical” experiments are used to calculate cloud susceptibilities (i.e., $-d\ln R_e/d\ln N_a$, $d\ln LWP/d\ln N_a$, $d\ln COD/d\ln N_a$, and $d\ln LCC/d\ln N_a$). Note that liquid cloud cover is not available from the CMIP6 simulations, so total cloud cover is used to calculate cloud cover response. Detailed information of CMIP6 simulations is given in Table 3.”

Table 3 (newly added). Experiment information for selected CMIP6 models: MPI-ESM-1-2-HAM and UKESM1-0-LL.

Model	MPI-ESM-1-2-HAM		UKESM1-0-LL	
Experiment ID	“hist-piAer”	“historical”	“hist-piAer”	“historical”
Activity ID	AerChemMIP	CMIP	AerChemMIP	CMIP
Institution ID	HAMMOZ- Consortium	HAMMOZ- Consortium	MOHC	MOHC
Nominal Resolution	250 km	250 km	250 km	250 km
Frequency	Monthly	Monthly	Monthly	Monthly
Variant label	r1i1p1f1	r1i1p1f1	r1i1p1f2	r1i1p1f2
[data version]	[20190627],	[20190627],	[20190813],	[20190406],

r2ilp1f1	r2ilp1f1	r2ilp1f2	r2ilp1f2
[20190627	[20190627],	[20191104],	[20190502],
/20190628]#,	r3ilp1f1	r3ilp1f2	r3ilp1f2
r3ilp1f1	[20190627	[20200128]	[20190502]
[20190627	/20191218]*		
/20191218]*			

#Note: data versions are 20190627 for Nd, Re, LWP and 20190628 for CC.

**Note: data versions are 20190627 for Nd, Re, LWP and 20191218 for CC.*

3) Figure 1a shows CESM2 and CNRM-ESM with similar ACI values (if I understand
correctly), yet Smith et al (2020) ([https://acp.copernicus.org/articles/20/9591/2020/acp-20-](https://acp.copernicus.org/articles/20/9591/2020/acp-20-9591-2020.html#&gid=1&pid=1)
[9591-2020.html#&gid=1&pid=1](https://acp.copernicus.org/articles/20/9591/2020/acp-20-9591-2020.html#&gid=1&pid=1)) reports stronger ACI for CESM2 in CMIP6 (Table 6). This
could due to many reasons, include the two major concerns that I raise above. So, it is good to
understand the reason as a sanity check for you results overall.

**Response:**

The ACI effects are referred to different things in Smith et al. (2020)⁸ and in this study. ACI in
Smith et al. (2020) refers to the radiative forcing induced by ACI, as correctly pointed out by
the reviewer that CEMS2’s ACI-induced shortwave radiative forcing is about three times larger
than the one from CNRM-ESM. While ACI in our study refers to the interactions themselves
between aerosol and cloud, i.e. changes in clouds due to aerosol.

We agree with the reviewer that this is a very good point. The difference between Smith et al.
(2020) and our study may point towards an important ACI uncertainty pathway in the GCMs.
To articulate this point, we have added a new discussion in *Discussion* part of the main text, as
below:

“Due to the complexity of clouds and climate systems, even a perfect simulation of cloud susceptibilities still cannot guarantee an unbiased estimate of ACI radiative forcing, this is because the efficiency in radiative effect of clouds varies significantly between GCMs; for example, CESM2 shows similar cloud susceptibilities to CNRM-ESM2-1 (Fig. 1a), but shows some of three times larger ACI radiative forcing⁸. A recent study highlighted that high equilibrium climate sensitivity (≥ 2.93 K) is needed to reproduce the trend in the observed Earth energy imbalance, where changes of clouds are critical for climate assessments but highly uncertain⁹. Therefore, radiative transfer processes of clouds and cloud feedback following ACI radiative forcing are also key to improving climate projections.”

**References:**

- 1. Malavelle FF, *et al.* Strong constraints on aerosol–cloud interactions from volcanic
eruptions. *Nature* **546**, 485-491 (2017).
- 2. Chen Y, *et al.* Machine learning reveals climate forcing from aerosols is dominated by
increased cloud cover. *Nature Geoscience* **15**, 609-614 (2022).
- 3. Oreopoulos L, Cho N, Lee D, Kato S. Radiative effects of global MODIS cloud regimes.
*Journal of Geophysical Research: Atmospheres* **121**, 2299-2317 (2016).
- 4. Neubauer D, *et al.* HAMMOZ-Consortium MPI-ESM1.2-HAM model output prepared
for CMIP6 CMIP historical. Earth System Grid Federation.
<https://doi.org/10.22033/ESGF/CMIP6.5016>. (2019).
- 5. Neubauer D, *et al.* HAMMOZ-Consortium MPI-ESM1.2-HAM model output prepared
for CMIP6 AerChemMIP hist-piAer. Earth System Grid Federation.
<https://doi.org/10.22033/ESGF/CMIP6.5007>. (2019).
- 6. Tang Y, *et al.* MOHC UKESM1.0-LL model output prepared for CMIP6 CMIP
historical. Earth System Grid Federation. <https://doi.org/10.22033/ESGF/CMIP6.6113>.
(2019).
- 7. O'Connor F. MOHC UKESM1.0-LL model output prepared for CMIP6 AerChemMIP
hist-piAer. Earth System Grid Federation. <https://doi.org/10.22033/ESGF/CMIP6.6062>.
(2019).
- 8. Smith CJ, *et al.* Effective radiative forcing and adjustments in CMIP6 models. *Atmos*
*Chem Phys* **20**, 9591-9618 (2020).
- 9. Myhre G, Hodnebrog Ø, Loeb N, Forster PM. Observed trend in Earth energy
imbalance may provide a constraint for low climate sensitivity models. *Science* **388**,
1210-1213 (2025).

**Response to Reviewer #3 (manuscript NCOMMS-25-29057):**

**General comments:**

Global climate models are unable to reproduce cloud cover response to aerosol. Y. Wang and
Co-Authors.

**Recommendation: Publish with revisions.**

The paper compares four measures of aerosol-cloud interaction (ACI) in 6 climate models (with
structural and parameter variations for one of them) with measures of these interactions inferred
from observed cloud responses to an effusive volcano eruption, which generated large sulfur
dioxide emissions. The resulting sulfate plume spread broadly over the North Atlantic, so
analyzing that area over a month covers a range of synoptic regimes to bolster the significance
of the results. The results provide an important evaluation of the ability of climate models to
simulate ACI. Publication is recommended. Suggested revisions follow.

**Response:**

We thank the reviewer for the positive feedback and constructive comments. All suggestions
have been carefully considered, and the manuscript has been revised accordingly. In particular,
we have added discussions on the prognostic cloud scheme in the results part and updated “data
and code availability”. Please find our detailed point-by-point responses below. A change-
tracked revision of the manuscript has also been uploaded.

**Revisions**

1) Four measures of ACI are presented. Except for the cloud cover susceptibility and the liquid
water path susceptibility for one of the models, all lie in or very close to the observed estimates.
Yet the title of the paper focuses on the single measure where the models fail. The paper should
be retitled to capture better the balance of results.

**Response:**

Thanks for the suggestion. Reflecting on the suggestions on the title from Reviewers-2&3, we
have changed to a new title: “*Challenges in global climate models to represent cloud response*
*to aerosol: insights from volcanic eruptions*”.

2) l. 117: LCC is presumably a fraction, so, unlike re and LWP which have distinct in-cloud
values, it is a grid property and should not be included with the others listed as in-cloud.

**Response:**

Agree, corrected as suggested.

3) 1. 138: The meaning of the phrase “comparing...observations” is not clear.

**Response:**

The sentence has been rephrased:

*“These overly strong LWP adjustments in GCMs compared to machine-learning-based*
*observational constraints¹ are in line with Malavelle et al.², who compared four GCMs with*
*climatological anomalies from satellite observations².”*

4) 1. 173: Do any of the models included in this study use a process-based, prognostic cloud
cover parameterization along the lines of Tiedkte (1989, Mon. Wea. Rev.), which is used, for
example, in GFDL CM4 (Zhao et al., 2018, 10.1002/2017MS001209; Held et al., 2019,
10.1029/2019MS001829)? It is possible that the prognostic scheme in ECHAM6.3-HAM2.3
is of this nature. In revision, discuss this among the structural aspects of the models.

**Response:**

The prognostic scheme in ECHAM6.3-HAM2.3³ is a Tiedkte-like scheme, which allows for
sub-saturation and supersaturation with respect to ice separately in the cloud-free and cloudy
air. So, it is along the lines of the scheme used in GFDL CM4 as you correctly mentioned. We
have added words to discuss the prognostic cloud schemes as suggested, see below.

*“Multi-scheme ACI structural uncertainty*

...

*In total, five different cloud schemes have been implemented in the ECHAM6.3-*
*HAM2.3 model to test various cloud microphysics and cover parameterisations. ... In*
*addition, we used the Prognostic cloud cover scheme (PROG_CC)³, a Tiedkte-like scheme⁴*
*similar to that in GFDL’s CM4.0 Climate Model^{5, 6}, which simulates cloud fraction*
*prognostically based on in-cloud water vapour and convective activity and uses an updated*
*cloud microphysics scheme. ...*

*As shown in Fig. 1b, all five cloud schemes in ECHAM6.3-HAM2.3 demonstrate a*
*reasonable simulation of the COD response. The reasonable COD response is again due to the*
*compensation of biases in the Twomey effect and the LWP adjustment. ... Nevertheless, none*
*of the cloud schemes reproduce the observed increase in cloud cover ($d\ln LCC/d\ln N_d$), even the*
*cloud cover schemes that depend on liquid water (XR) or are prognostic (PROG_CC) still*
*underestimate $d\ln LCC/d\ln N_d$, suggesting a major structural uncertainty in current cloud cover*
*schemes. Further investigations of the cloud cover schemes and relevant cloud processes are*
*imperative to clarify cloud cover response to aerosol changes and to inform model*
*improvements.”*

5) ll. 238-240: Nordling et al. (2024, 10.5194/acp-24-869-2024) is a parameterization of the
type described here and should be referenced.

**Response:**

Thanks for suggesting this interesting article, we have referenced it.

Updated sentences:

*“A combination of machine learning and the results of high-resolution cloud-resolving models*
*would be a plausible way forward for improving cloud parameterisations in global models to*
*better represent the cloud cover’s responses to aerosol changes⁷. It is important to note that*
*clouds respond differently to aerosol changes under various meteorological conditions^{8,9} (e.g.,*
*precipitation-dominant vs entrainment-dominant stratocumulus regimes) and background*
*aerosol levels¹⁰, so more observational constraints in diverse conditions are critical to validate*
*and calibrate cloud sensitivities in future.”*

6) The data availability section includes ECHAM-HAMMOZ but not the other models.

**Response:**

We have uploaded GCM simulation data used to produce the results in this study to Zenodo,
which will be published with the manuscript. We have updated “Data Availability” and “Code
Availability”, see below.

**“Data Availability**

*The GCM simulation data used to produce the results will be available at Zenodo*
*(<https://doi.org/10.5281/zenodo.16926288>). The MODIS cloud products from Aqua*
*(MYD08_L3) and Terra (MOD08_L3) are openly available from the Atmosphere Archive and*
*Distribution System Distributed Active Archive Center of National Aeronautics and Space*
*Administration (LAADS-DAAC, NASA) (<https://ladsweb.modaps.eosdis.nasa.gov>). CMIP6*
*data for MPI-ESM-1-2-HAM and UKESM1-0-LL models are openly available through the*
*Earth System Grid Federation (ESGF) network (<https://esgf-ui.ceda.ac.uk/cog/projects/esgf->*
*[ceda/](https://esgf-ui.ceda.ac.uk/cog/projects/esgf-)).*”

**“Code Availability**

*The ECHAM-HAMMOZ model (ECHAM6.3-HAM2.3 and ECHAM6.3-SALSA2.0) is available*
*to the scientific community under the HAMMOZ Software License Agreement at*
*https://redmine.hammoz.ethz.ch/projects/hammoz/wiki/2_How_to_get_the_sources. The*
*UKESM1 is released for use by UK researchers at <https://ukesm.ac.uk/model-releases/>. The*
*CESM2 is available at: <https://www.cesm.ucar.edu/models/cesm2/download>. The documents*
*and code of each module of CNRM-ESM2-1 are documented on the CNRM website*

(<https://www.umr-cnrm.fr/cmip6/>). CAM5.3-Oslo (the atmospheric and aerosol module of
NorESM) code is available from GitHub: <https://github.com/NorESMhub/>.”

**References:**

- 1. Chen Y, *et al.* Machine learning reveals climate forcing from aerosols is dominated by
increased cloud cover. *Nature Geoscience* **15**, 609-614 (2022).
- 2. Malavelle FF, *et al.* Strong constraints on aerosol–cloud interactions from volcanic
eruptions. *Nature* **546**, 485-491 (2017).
- 3. Muench S, Lohmann U. Developing a Cloud Scheme With Prognostic Cloud Fraction
and Two Moment Microphysics for ECHAM-HAM. *Journal of Advances in Modeling*
*Earth Systems* **12**, e2019MS001824 (2020).
- 4. Tiedtke M. Representation of Clouds in Large-Scale Models. *Monthly Weather*
*Review* **121**, 3040-3061 (1993).
- 5. Zhao M, *et al.* The GFDL Global Atmosphere and Land Model AM4.0/LM4.0: 2.
Model Description, Sensitivity Studies, and Tuning Strategies. *Journal of Advances in*
*Modeling Earth Systems* **10**, 735-769 (2018).
- 6. Held IM, *et al.* Structure and Performance of GFDL's CM4.0 Climate Model. *Journal*
*of Advances in Modeling Earth Systems* **11**, 3691-3727 (2019).
- 7. Nordling K, *et al.* Technical note: Emulation of a large-eddy simulator for
stratocumulus clouds in a general circulation model. *Atmos Chem Phys* **24**, 869-890
(2024).
- 8. Prabhakaran P, Hoffmann F, Feingold G. Evaluation of Pulse Aerosol Forcing on
Marine Stratocumulus Clouds in the Context of Marine Cloud Brightening. *Journal of*
*the Atmospheric Sciences* **80**, 1585-1604 (2023).
- 9. Chen Y, *et al.* Substantial cooling effect from aerosol-induced increase in tropical
marine cloud cover. *Nature Geoscience* **17**, 404-410 (2024).
- 10. Carslaw KS, *et al.* Large contribution of natural aerosols to uncertainty in indirect
forcing. *Nature* **503**, 67-71 (2013).

**Response to Reviewer #1 (manuscript NCOMMS-25-29057A):**

**General comments:**

Review of Challenges in global climate models to represent cloud response to aerosol: insights
from volcanic eruptions. By Wang et al.

I have read over the responses to reviews, looked at some of the background literature and read
through the revised paper. The main point of the paper is that models simulate the AOD change
associated with the 2014 effusive eruption well, but do not get the apportionment between Re,
LWP and CF correct, mainly because they do not predict a CF increase in response to the
aerosols. The latter point is based on an acceptance of the Chen et al. (2022) paper, in which
machine learning is used to predict these quantities as measured by MODIS, using
meteorological data from ER5. I am not sure this result has been independently verified.
Monthly mean data for October are used for the ML model, so about 19 Octobers in total,
although I assume many 1-degree grid boxes. I am left feeling a bit skeptical about whether the
Chen et al. (2022) residual estimates of the susceptibilities are robust enough to tune models
to. I think this dependence should be a bit more emphasized in the paper. The paper makes it
seem like the satellite data are rock solid, but there are major uncertainties in both the
observational apportionment and its fitting with the ML method and these things are critical to
the conclusions. It seems a bit worrisome that straightforward comparison of climatologies
reveals no change in CF in 2014, but using the ML residual method introduces one.

Abstract and Conclusions: This work concludes that models underestimate cloud amount
changes compared to MODIS, but previous work with MODIS data by McCoy and
Hartmann(2015) Malavelle et al (2017) see large Twomey effect but little change in water path
or fraction. Malavelle et al(2017) use MODIS data and state that “cloud amount changes are
undetectable” What changed in the interim to make cloud amount changes a significant effect?
This seems to be based on the work of Wang et al. (2022) that use ML modeling of October
monthly mean data to estimate the meteorological effects on cloud properties. Although
straightforward average comparisons suggest no change in LWP or CF between September or
October of 2014 with the long-term climatology for those months, the ML method finds an
increase in cloud fraction in October 2014 in response to aerosols. This seems to suggest that
the ML model predicted a decrease in CF in October 2014 that was not observed and then
attributed the difference to the aerosols. I have not studied the methods thoroughly, but I am
somewhat unconvinced, particularly since the September data were not used, which would
have given an independent data point. I find the simple result that no significant difference in
cloud fraction was observed in 2014 more compelling than the residual of a ML computation
based on monthly data (20 months total) and the residual derived therefrom. What am I missing?

**Response:**

Thank reviewer for further clarification of your concern. We understand why you feel skeptical
about the increase of cloud fraction (CF) in Chen et al. (2022) using machine-learning (ML),
while Malavelle et al (2017) found “cloud amount changes are undetectable” using
climatological anomaly. We will clearly explain this with multiple lines of evidence, shown as
below.

**i) Discrepancy between Malavelle et al. (2017) and Chen et al. (2022)**

In Malavelle et al. (2017), their analysis of cloud fraction was hampered by the use of
Collection-5 (C5) MODIS-Aqua only data from 2002-2014, which in retrospect should have
not been used for cloud fraction analysis.

Cited from Page-98 in the MODIS User Guide (Platnick et al., 2018):

“In C5, those pixels restored to clear by the Clear Sky Restoral algorithm (i.e., those with flag CSR=2) were inadvertently excluded from the denominator in the calculation of fraction. In C6, the retrieval fraction is now correctly defined ...”

51
We have performed climatological anomaly with the corrected MODIS Collection-6 cloud
fraction (same approach as used in (Malavelle et al., 2017)), a strong increase of CF can also
be clearly seen across the whole north Atlantic (see Figure below).

[REDACTED]

Extended Data Fig. 3c from Chen et al. (2022) | Climatological anomaly of CF
observed by MODIS-Aqua in October 2014.

This consistence of the increase of CF in both ML and climatological anomaly should already
ease the concern of reviewer. In Chen et al. (2022), they have explicitly noted the caveat in CF
in C5 and the correction of the latest C6 version (text provided below).

Cited from Method section in Chen et al. (2022):

“An inadvertent artifact in the calculations of cloud fraction (derived from Cloud Optical Property) in Collection 5.1 has also been removed in

*Collection 6.1 (Platnick et al., 2018) and both Terra-MODIS and Aqua-*
*MODIS now show consistent results (Platnick et al., 2017; Hubanks et al.,*
*2019).”*

The difference in ΔCF analysis when using ML and climatological anomaly approaches has
been discussed in detail in Chen et al. (2022) Supplementary Information Section-S2. We do
not include this discussion or clarification in our manuscript, because our study is a modelling
study rather than satellite observation analysis, and because this information has already been
provided in Chen et al. (2022) and we feel it is not necessary to duplicate in this manuscript.
We appreciate there is still uncertainty in the satellite observations, just like all kinds of
observations have. Following reviewer suggestion, we have acknowledged the uncertainty and
limitation of this study in the manuscript, revised text provided in the Point-3 response.

We would also like to stress that, although artifact in CF calculation in the MODIS-C5 data,
liquid water path (LWP) retrieval in C5 is still reliable, Chen et al. (2022) find consistent nearly
zero response in LWP as in Malavelle et al. (2017); therefore, the main conclusion of Malavelle
et al. (2017) of weak LWP response is still validated.

**ii) Independent data points as evidence: September 2014 and Hawaii volcanoes.**

As suggested by the reviewer, September 2014 Holuhraun eruption could be an independent
data point to further validate the increase of CF from ML. This analysis has been performed in
Chen et al. (2022). Briefly, September 2014 also shows a large increase of CF in both ML and
climatological anomaly approaches (see figure below), which is in line with Oct. 2014 case;
see detailed discussion in Chen et al. (2022) -- Supplementary Information Section-S1. Here,
we would like to focus on Oct. 2014, this is in line with Malavelle et al. (2017) who also focus
on Oct. 2014 and only some Sep. 2014 discussion in their supplement; this is because October
is more representative of the global distribution of cloud regimes due to more limited areal
extent of the plume in September, and because noise from continental pollution in lower part
of the domain in September (Malavelle et al., 2017).

[REDACTED]

Extended Data Fig. 9 in Chen et al. (2022) | Cloud fraction (CF) responses to Holuhraun volcanic aerosol in September 2014.

In addition, we would like to stress again the degassing eruptions of Kilauea volcano in Hawaii, also provide independent data point to prove the fidelity of ML approach and increase of CF. As we provided in the first-round review, Chen et al. (2024) applied the same machine-learning approach to the trade wind regions of Hawaii-volcanoes; they find no ACI signal in the upwind clean region but a strong ACI signal only in the downwind region polluted by volcanic aerosols (see Fig. R1). This strongly validates the fidelity of ML approach for detecting CF increase.

Figure R1. Cloud susceptibility to volcanic aerosol: (a) downstream polluted region; (b) upstream clean region. Ratio = MODIS / ML-MODIS. Black boxplots are the validation against non-eruption years; red boxplots are the detected ACI signals. Monto-Carlo 648 ensembles are analysed. Source: figure is adapted from Figs. 2 and S2 in Chen et al. (2024).

In summary, multiple lines of evidence all suggest that ML detected increase of CF is robust, including i) the consistent increase of CF using both ML and climatological approach (same approach as in Malavelle et al. (2017)), ii) strong increase of CF in September 2014 from Holuhraun volcanic aerosol, iii) strong increase of CF in downstream regions of Kilauea aerosol plume but insignificant change of CF in upstream clean region.

**Point-by-point response to specific comments:**

1) Abstract concludes with “, and support Net-Zero Policymaking” makes it seem like the
purpose of scientific research is to support a particular policy option. I strongly recommend
against using this phrasing.

**Response:**

*Removed this statement.*

2) 45-46 “none of state-of-the-art climate models are able to fully account for this recent
warming”. This statement is hyperbolic and makes it seem like we don’t know what we are
doing, and it also is unbalanced in a similar way to people who said warming had stopped after
1998. There was a big jump in 2023 from 2022, but equally remarkable is the small amount of
warming from 1998 to 2014. Smoothing over these anomalies, the overall trend has continued
apace at about 0.2K/decade, which is what the models are predicting (if this trend is extrapolated
from 2024 to 2100 it reaches about 3K anomaly in 2100, in good agreement with central
estimates for a scenario that does not rapidly reduce CO2 emissions.). One should not expect
models to exactly explain one year’s anomaly because of the major influences of natural
variability and the uneven, sometimes stepwise response to human influences. The statement
that models cannot “fully account” for the anomaly in 2023 is probably strictly true for every
109 year in the record. It is not helpful to make things seem more uncertain than they probably are.

**Response:**

*Removed this sentence.*

3) 82 “both positive and negative results reported” Also a lot of zero effect on LWP and CF.
Can also depend on meteorology and the magnitude of the aerosol source compared to
background. Evidence that background levels decrease the effect of emissions of sulfate and
the effect is also not linear, diminishing for very large sulfate loading. Therefore one case like
Iceland cannot be the silver bullet for the global problem.

**Response:**

*Agree, sorry for not being rigorous in wording. Rephrase to “with signals in the range from*
*positive to negative reported”. We have also added the discussion of dependence on the*
*meteorology and aerosol source strength compared to background, as well as acknowledge that*
*one case like Iceland volcano cannot solve all problem and studies of more opportunistic*
*experiments are needed, such as IMO shipping emission reduction since 2020 (Gettelman et*
*al., 2024). The revised text read as below:*

“Our study is not without limitation. The observational constraints may suffer from uncertainty in satellite observations, despite that satellites are currently the only measure to provide continuous long-term cloud cover observations over a

large scale. In future studies, using multiple satellites along with machine-learning development could further reduce uncertainty in the observations of cloud cover and cloud susceptibility to better constrain global climate models. The nonlinearity of cloud response to aerosol means that cloud susceptibilities can also depend on meteorology (Gettelman et al., 2015; Chen et al., 2024) and the strength of aerosol source compared to background (Gettelman et al., 2015; Carslaw et al., 2013). For example, Gettelman et al. (2015) performed sensitivity simulations of Holuhraun-2014 using a GCM and demonstrated that meteorological diversity could introduce up to 30% difference in ACI radiative forcing, and that the forcing is not proportional to the change in aerosol emissions. This suggests the need of more opportunistic experiment studies, such as IMO shipping emission reduction since 2020 (Gettelman et al., 2024), together with the Holuhraun-2014 to have a holistic investigation of ACI in diverse global conditions.”

4) 95-96 Are there references for this statement that the observational uncertainty has been
removed? Also, explain why the observational response of CF is now large, compared to earlier
observational work that suggested it is small. I would be skeptical if it is based on AI
methodologies that do not have a strong physical basis and explanation. The LWC and CF
changes do now show up in simple averages, only the Twomey effect.

**Response:**

Aggregating large observational samples can minimize the uncertainty, this concept has been
well accepted and practised in many disciplines, and this is why large sample size is always
recommended for robust studies in almost all disciplines although need to compromise with
the required resources (Schillaci and Schillaci, 2022). Specific to MODIS, for example, (Wei
et al., 2020) demonstrate that by aggregating relatively larger sample size in 10 km resolution
shows higher accuracy than 3 km resolution data. In Chen et al. (2022), they looked into and
aggregate CF over a region of about 3000 km × 6000 km (over 1000 samples), suggesting a
large reduce of random uncertainty. Again, if the detected CF signal was dominated by the
satellite uncertainty, can we expect signals as shown in Fig. R1? Absolutely, not. Because we
would expect similar signals in both downwind and upwind region. However, what we see is:
very large ACI signals only in the volcanic aerosol plume downwind region (Fig. R1a)
contrasting with near zero signal in the upwind clean region (Fig. R1b).

Yes, the ML approach used in Chen et al. (2022) does not have a strong physical basis neither
they claimed physical basis, and this is exactly why we need to use and improve physical-based
models (such as this study) to understand the phenomenon observed in Chen et al. (2022). No
strong physical basis does not hamper ACI signal detection in Chen et al. (2022). The reviewer
state that (s)he did not study Chen et al. (2022) method thoroughly, below we try to explain

their method in plain language as best as we can. In Chen et al. (2022), they use ML with
extensive meteorological parameters as inputs to build the counterfactual clouds for given
meteorology, i.e. ML-MODIS. Then, ML-MODIS for 2014 could be understood as the
counterfactual “experimental” case, and ML-MODIS for other years without volcano as the
counterfactual “control” cases. The “control” cases are well validated against MODIS
observations (e.g., black boxplots in Fig. R1a -- lying on 1.0 line means well validation) with
hundreds Monte-Carlo ensembles; however, the “experimental” case show a large “gap”
between ML-MODIS and MODIS (e.g. red boxplots in Fig. R1a). This means the observed
“gap” in the “experimental” case cannot be explained by the meteorology, therefore rejecting
the null hypothesis: this “gap” is due to meteorology. The major difference between “control”
and “experimental” is the volcanic aerosol, therefore, the “gap” needs to be explained by
volcanic ACI. This ML approach overcomes the challenge in climatological anomaly approach
-- we do not know if the anomaly is due to aerosol or meteorology (since meteorology in 2014
is not identical to the climatological average). Hope this is helpful for understanding the logic
of ACI signal detection in Chen et al. (2022), where physical basis is not the key.

Regarding the discrepancy in CF response, as we explained above, it is because the artifact in
the calculations of cloud fraction in Collection 5.1 (Platnick et al., 2018), which in retrospect
should have not been used for cloud fraction analysis.

5) 101 – “we suggest that the wider climate community takes the Holuhraun-2014 eruption as
an anchor case” Several papers have already investigated the eruptions effect on clouds and
compared with models, so what is being suggested is not new. Malavelle et al. (2017)
Christensen et al. (2022), Marshall et al.(2022), Chen et al. (2022), Jordan et al.(2024),
Feingold et al.(2024), Chen et al.(2024, Hawaiian Volcanoes), Peace et al.(2024), Zoega et
al.(2025a), Zoega et al.(2025b), Marelle et al. (2025).

**Response:**

What is new here is that models constraining needs to consider not only state variables (such
as how much cloud fraction) but also cloud susceptibilities (especially for cloud fraction), as
we highlighted in the beginning of the manuscript: “*Cloud susceptibility is critical for climate
projections, large discrepancies in cloud susceptibility can propagate significant
uncertainties into climate projections, even with similar current states of clouds and
aerosols ...*”.

Although we have stressed this new point in the Results section: “*In contrast to the widely used
method of evaluating cloud and aerosol states, here we re-evaluate the performance of GCMs
using cloud susceptibilities ...*”, we agree that this new point is not emphasized in this sentence.
We therefore have rephrased the sentence to credit Malavelle et al. (2017) effort in this Iceland
case, and emphasized again this new point in cloud susceptibility, read as:

“We echo Malavelle et al. (2017) to call for the wider climate community takes the Holuhraun-2014 eruption as an anchor case, and stress validation of cloud susceptibilities in addition to cloud properties to decompose and unravel the largest uncertainty in climate radiative forcing, i.e. ACI simulation ...”

6) 156-157: Getting the albedo change right is central. It is also much more accurately measured
that apportionment of albedo change to Re, LWP and CF. I wonder if it can be shown that the
observational apportionment between Re, LWP and CF is less uncertain than that in the models.
Based on my reading of the Chen et al. (2022) paper from which the ML estimates come, I am
not sure, Is the ML method used here good enough to use to change models?

**Response:**

Indeed, to understand how much uncertainty in the models, we will need observations. The
ACI signals from Chen et al. (2022) is robust, as we explained in the first Response, where we
have explained the discrepancy between Chen et al. (2022) and previous study and provided
independent data points following reviewer’s suggestion.

**References:**

Carslaw, K. S., Lee, L. A., Reddington, C. L., Pringle, K. J., Rap, A., Forster, P. M., Mann, G.
206 W., Spracklen, D. V., Woodhouse, M. T., Regayre, L. A., and Pierce, J. R.: Large
contribution of natural aerosols to uncertainty in indirect forcing, *Nature*, 503, 67,
10.1038/nature12674, 2013.

Chen, Y., Haywood, J., Wang, Y., Malavelle, F., Jordan, G., Peace, A., Partridge, D. G., Cho,
210 N., Oreopoulos, L., Grosvenor, D., Field, P., Allan, R. P., and Lohmann, U.: Substantial
cooling effect from aerosol-induced increase in tropical marine cloud cover, *Nature*
*Geoscience*, 10.1038/s41561-024-01427-z, 2024.

Chen, Y., Haywood, J., Wang, Y., Malavelle, F., Jordan, G., Partridge, D., Fieldsend, J., De
Leeuw, J., Schmidt, A., Cho, N., Oreopoulos, L., Platnick, S., Grosvenor, D., Field, P.,
and Lohmann, U.: Machine learning reveals climate forcing from aerosols is dominated
by increased cloud cover, *Nature Geoscience*, [https://doi.org/10.1038/s41561-022-](https://doi.org/10.1038/s41561-022-00991-6)
[00991-6](https://doi.org/10.1038/s41561-022-00991-6), 2022.

Gettelman, A., Schmidt, A., and Egill Kristjánsson, J.: Icelandic volcanic emissions and climate,
*Nature Geoscience*, 8, 243-243, 10.1038/ngeo2376, 2015.

Gettelman, A., Christensen, M. W., Diamond, M. S., Gryspeerdt, E., Manshausen, P., Stier, P.,
Watson-Parris, D., Yang, M., Yoshioka, M., and Yuan, T.: Has Reducing Ship Emissions
Brought Forward Global Warming?, *Geophysical Research Letters*, 51,
e2024GL109077, <https://doi.org/10.1029/2024GL109077>, 2024.

Hubanks, P., Platnick, A. S., King, M., and Ridgway, B.: MODIS Atmosphere L3 Gridded
Product Algorithm Theoretical Basis Document (ATBD) & Users Guide, available from:
<https://icdc.cen.uni->

hamburg.de/fileadmin/user_upload/icdc_Dokumente/MODIS/MODIS_Collection6_A
 [atmosphereL3 GriddedProduct ATBDandUsersGuide v4.1 Sep22 2015.pdf](https://atmosphereL3.GriddedProduct_ATBDandUsersGuide_v4.1_Sep22_2015.pdf), 2019.

Malavelle, F. F., Haywood, J. M., Jones, A., Gettelman, A., Clarisse, L., Bauduin, S., Allan, R.
 P., Karset, I. H. H., Kristjánsson, J. E., Oreopoulos, L., Cho, N., Lee, D., Bellouin, N.,
 Boucher, O., Grosvenor, D. P., Carslaw, K. S., Dhomse, S., Mann, G. W., Schmidt, A.,
 Coe, H., Hartley, M. E., Dalvi, M., Hill, A. A., Johnson, B. T., Johnson, C. E., Knight,
 233 J. R., O'Connor, F. M., Partridge, D. G., Stier, P., Myhre, G., Platnick, S., Stephens, G.
 234 L., Takahashi, H., and Thordarson, T.: Strong constraints on aerosol–cloud interactions
 from volcanic eruptions, *Nature*, 546, 485-491, 10.1038/nature22974, 2017.

Platnick, A. S., Meyer, G. K., King, D. M., Wind, G., Amarasinghe, N., Marchant, B., Arnold,
 G. T., Zhang, Z., Hubanks, A. P., Ridgway, B., and Riedi, J.: MODIS Cloud Optical
 Properties: User Guide for the Collection 6/6.1 Level-2 MOD06/MYD06 Product and
 Associated Level-3 Datasets, [https://atmosphere-](https://atmosphere-imager.gsfc.nasa.gov/sites/default/files/ModAtmo/MODISCloudOpticalPropertyUserGuideFinal_v1.1_1.pdf)
 [imager.gsfc.nasa.gov/sites/default/files/ModAtmo/MODISCloudOpticalPropertyUser](https://atmosphere-imager.gsfc.nasa.gov/sites/default/files/ModAtmo/MODISCloudOpticalPropertyUserGuideFinal_v1.1_1.pdf)
 [GuideFinal_v1.1_1.pdf](https://atmosphere-imager.gsfc.nasa.gov/sites/default/files/ModAtmo/MODISCloudOpticalPropertyUserGuideFinal_v1.1_1.pdf), 2018.

Platnick, S., Meyer, K. G., King, M. D., Wind, G., Amarasinghe, N., Marchant, B., Arnold, G.
 243 T., Zhang, Z., Hubanks, P. A., Holz, R. E., Yang, P., Ridgway, W. L., and Riedi, J.: The
 244 MODIS Cloud Optical and Microphysical Products: Collection 6 Updates and
 245 Examples From Terra and Aqua, *IEEE Transactions on Geoscience and Remote*
 *Sensing*, 55, 502-525, 10.1109/TGRS.2016.2610522, 2017.

Schillaci, M. A. and Schillaci, M. E.: Estimating the population variance, standard deviation,
 and coefficient of variation: Sample size and accuracy, *Journal of Human Evolution*,
 171, 103230, <https://doi.org/10.1016/j.jhevol.2022.103230>, 2022.

Wei, J., Li, Z., Sun, L., Peng, Y., Liu, L., He, L., Qin, W., and Cribb, M.: MODIS Collection
 6.1 3 km resolution aerosol optical depth product: global evaluation and uncertainty
 analysis, *Atmospheric Environment*, 240, 117768,
 <https://doi.org/10.1016/j.atmosenv.2020.117768>, 2020.

**Response to Reviewer #2 (manuscript NCOMMS-25-29057A):**

My comments have been addressed appropriately, and I recommend acceptance of the
manuscript.

**Response:**

We thank the reviewer for positive evaluation and the constructive comments, which helped
improved this study greatly.

Review

Global climate models are unable to reproduce cloud cover response to aerosol. Y. Wang and Co-Authors.

Recommendation: Publish with revisions.

The paper compares four measures of aerosol-cloud interaction (ACI) in 6 climate models (with structural and parameter variations for one of them) with measures of these interactions inferred from observed cloud responses to an effusive volcano eruption, which generated large sulfur dioxide emissions. The resulting sulfate plume spread broadly over the North Atlantic, so analyzing that area over a month covers a range of synoptic regimes to bolster the significance of the results. The results provide an important evaluation of the ability of climate models to simulate ACI. Publication is recommended. Suggested revisions follow.

Revisions

1. Four measures of ACI are presented. Except for the cloud cover susceptibility and the liquid water path susceptibility for one of the models, all lie in or very close to the observed estimates. Yet the title of the paper focuses on the single measure where the models fail. The paper should be retitled to capture better the balance of results.
2. l. 117: LCC is presumably a fraction, so, unlike r_e and LWP which have distinct in-cloud values, it is a grid property and should not be included with the others listed as in-cloud.
3. l. 138: The meaning of the phrase “comparing...observations” is not clear.
4. l. 173: Do any of the models included in this study use a process-based, prognostic cloud cover parameterization along the lines of Tiedtke (1989, *Mon. Wea. Rev.*), which is used, for example, in GFDL CM4 (Zhao et al., 2018, [10.1002/2017MS001209](https://doi.org/10.1002/2017MS001209); Held et al., 2019,

[10.1029/2019MS001829](https://doi.org/10.1029/2019MS001829)? It is possible that the prognostic scheme in ECHAM6.3-HAM2.3 is of this nature. In revision, discuss this among the structural aspects of the models.

5. II. 238-240: Nordling et al. (2024, 10.5194/acp-24-869-2024) is a parameterization of the type described here and should be referenced.
6. The data availability section includes ECHAM-HAM0Z but not the other models.